



# A Survey of Radiative and Physical Properties of North Atlantic Mesoscale Cloud Morphologies from Multiple Identification Methodologies

Ryan Eastman [1], Isabel L. McCoy [2, 3, 4], Hauke Schulz [1, 5], and Robert Wood [1]

[1]Dept. of Atmospheric Sciences, University of Washington, Seattle, WA, USA
[2]Cooperative Institute for Research in Environmental Sciences, University of Colorado, Boulder, CO, USA
[3]National Oceanic and Atmospheric Administration, Chemical Sciences Laboratory, Boulder, CO, USA
[4]Cooperative Programs for the Advancement of Earth System Science, University Corporation for Atmospheric Research, Boulder, CO, USA
[5]Cooperative Institute for Climate, Ocean and Ecosystem Studies, University of Washington, Seattle, WA, United States

**Correspondence:** Ryan Eastman (rmeast@uw.edu)

**Abstract.**

Three supervised neural network cloud classification routines are applied to daytime MODIS Aqua imagery and compared for the year 2018 over the North Atlantic Ocean: The Morphology Identification Data Aggregated over the Satellite-era (MIDAS), which specializes in subtropical stratocumulus (Sc) clouds; Sugar, Gravel, Flowers, and Fish (SGFF), which is focused on shallow cloud systems in the tropical trade winds; and the community record of marine low-cloud mesoscale morphology supported by the NASA Making Earth Science Data Records for Use in Research Environments (MEaSUREs) dataset, which is focused on shallow clouds globally.

Comparisons of co-occurrence and vertical and geographic distribution show that morphologies are classified in geographically distinct regions: shallow suppressed and deeper aggregated and disorganized cumulus are seen in the tropical trade winds. Shallow Sc types are frequent in subtropical subsidence regions. More vertically developed solid stratus and open and closed cell Sc are frequent in the mid-latitude storm track. Differing classifier routines favor noticeably different distributions of equivalent types.

Average scene albedo is more strongly correlated with cloud albedo than cloud amount for each morphology. Cloud albedo is strongly correlated with the fraction of optically thin cloud cover. The albedo of each morphology is dependent on latitude and location in the mean anticyclonic wind flow over the N. Atlantic. Strong rain rates are associated with middling values of albedo for many cumuliform types, hinting at a complex relationship between the presence of heavily precipitating cores and cloud albedo. The presence of ice at cloud top is associated with higher albedos. For a constant albedo, each morphology displays a distinct set of physical characteristics.





## 1 Introduction

Low clouds tend to organize into large-scale, repeating morphological structures with individual cells observed on the scale of 20-150 km in patterns that repeat for hundreds or even thousands of kilometers (Agee, 1987; Muhlbauer et al., 2014). These structures influence climate in different ways due to their unique radiative characteristics (McCoy et al., 2023) and sensitivities to their surrounding environment (Qu et al., 2015). Understanding where and how different morphological structures develop, and what the radiative characteristics of those structures are is vital for understanding how low clouds will evolve with climate

change and for determining the sensitivity of our climate.

Clouds can evolve between morphologies via multiple pathways depending on initial environmental conditions and subsequent changes to those environmental conditions (Bretherton et al., 2010; Yamaguchi et al., 2017; Eastman et al., 2022; Salazar and Tziperman, 2023). Additionally, differing cloud morphologies can experience opposite changes when exposed to the same environmental forcing. For instance stratiform clouds, which form beneath temperature inversions and are driven by radiative

cooling at cloud top, will reduce in extent when a warming sea surface weakens the inversion. However, the warming sea surface will drive more upward motion within the boundary layer, causing cumulus (Cu) to replace stratus (St). This process is detailed in Wyant et al. (1997) and is also shown in Norris et al. (1998) and Eastman et al. (2011). This process shows how one archetypal cloud organization (e.g., Stratocumulus, Sc) can be replaced by another (e.g., Cu) when environmental conditions (sea surface temperature, SST) changes, and is one example of many possible changes in cloud organization associated with a

changing climate.

Until recently, surface observations were the primary source of cloud type information, including the studies referenced above. Trained observers classify cloud types at multiple levels as part of coordinated weather reporting (WMO, 1974), and these classifications have contributed to long-term climate records (Hahn et al., 2009). These records have been valuable assets in studying long-term cloud and climate behaviors (Klein et al., 1995; Norris et al., 1998; Eastman et al., 2011), but are limited

in their spatial resolution and are prone to incongruities in their record due to geopolitical and economic shifts (Warren et al., 1991). Satellite-based cloud-type data are now being developed in an attempt to continue and enhance the study of cloud types. Several methods for systematically identifying low cloud morphological structure have recently been developed (Wood and Hartmann, 2006; Rasp et al., 2020; Yuan et al., 2020; Denby, 2020; Janssens et al., 2021). This development coincides with advances in the spatial and spectral resolution of satellite observations along with exponentially improved computing power.

The three routines compared here are human-trained or supervised machine learning algorithms that classify cloud cover using satellite images. First, human observers classify morphological structures on hundreds or thousands of satellite images. These classifications are then used to train a neural network, which can then identify these specific structures on other images from the same sensor. These classifying routines have some flexibility as they can be run on all available data or applied to other data sources. Aside from human-trained algorithms, routines are being developed that identify and sort morphologies

without initial training (i.e., unsupervised, Denby, 2020). Future work may focus on comparing unsupervised classifications with those from supervised methods.



Cloud classifiers have been developed to identify archetypal cloud morphologies for a variety of climatological regions. The Morphology Identification Data Aggregated over the Satellite-era (MIDAS, Wood and Hartmann (2006); updated in McCoy et al. (2023)) dataset was trained to discern between open and closed cell Sc fields in subtropical subsidence regions, also pro-
ducing a "disorganized, but cellular" category representing any remaining cloud cover that has cellular structure. The Sugar, Gravel, Flowers, Fish (SGFF, Schulz et al., 2021) algorithm was trained in the North Atlantic trade wind regime, and identifies four cloud morphologies more common to the tropics and has limited overlap with the MIDAS dataset (e.g., mostly the disorganized type, Rasp et al., 2020). The community record of marine low-cloud mesoscale morphology supported by the NASA Making Earth Science Data Records for Use in Research Environments (MEaSUREs, Yuan et al., 2020) routine produced a
more geographically varied dataset by training the algorithm with images sourced globally, and with cloud morphologies ranging from solid marine stratus to clustered tropical convection. A focus of this work is to understand the extent to which these algorithms classify the same patterns and variability, despite their differing training routines and areas. This is still an open question and has important implications for how we place studies based on these varied routines into context with one another.

Prior work has shown that cloud albedo is a function of both cloud amount and morphology globally (e.g., McCoy et al.,
2017, 2023). McCoy et al. (2023) found that the relationship between scene albedo and cloud amount is significantly different depending on cloud morphology, with closed-cell Sc clouds reflecting more than open cell Sc or disorganized Sc for the same cloud amount. They found this was in-part due to the different fractions of optically thin cloud cover between morphologies, clearly illustrating how cloud amount alone does not fully explain cloud albedo. The three MIDAS cloud types, which are especially skilled for open or closed cell Sc identification, were utilized in this analysis. This motivates further evaluation of
this behavior using more specific, largely tropical cloud type identifications to subset the expansive disorganized category. The global focus of McCoy et al. (2023) also motivates evaluating behaviors on a more regional scale to better understand their variability since cloud micro- and macro-physical characteristics and radiative properties may be affected by geographic location.

This work will assess and compare geographic, radiative, and physical differences for a variety of cloud types identified by
the supervised neural network algorithms discussed above (i.e., MIDAS, SGFF, and MEASURES) for one year in the North Atlantic. The characteristics of our three classification routines can be compared across several climate regimes in the N. Atlantic, from the mid-latitude storm track to subtropical subsidence regions and the tropical trade-winds. Cloud properties within each routine will also be compared against one another. In particular, we seek to quantify the contributions that a varied range of cloud morphologies make to the global cloud amount-albedo relationships and further investigate the albedo sensitivity
to variations in cloud characteristics across morphology types.

## 2  Data

Data in this manuscript span the entire year 2018 for the North Atlantic, defined by a box bounded by 0-90° W and 5-55° N. Only ocean areas are considered in this work. The region and time were chosen because classifier data from all three sources were reliably available for that entire year in that region, and also because the North Atlantic Ocean contains a wide variety of



climatological conditions in a single ocean basin, including a strong mid-latitude storm track in the north, a subtropical subsidence region in the east, and tropical trade winds to the south. Classifier data will soon be available for more regions and more dates, allowing for more extensive studies of morphologies. Data from all three classifier routines come from the Moderate Resolution Imagine Spectroradiometer (MODIS) radiometer on the polar orbiting Aqua satellite, which crosses the equator at 01:30 and 13:30 local time (LT). For this work, only the daytime swaths are used. In order to better compare datasets built by these differing routines at differing resolutions, all morphology data are projected onto a $1° \times 1°$ latitude/longitude grid. Each $1° \times 1°$ box is classified as a morphology if any part of that box was classified by a routine. This means that a $1° \times 1°$ box can be classified multiple times by the same classifier if multiple cloud morphologies are observed in that box. This allows for the study of overlap, where certain boxes may be located in a transitioning regime between two morphologies.

## 2.1 Classifier Routines

### 2.1.1 MIDAS

The Morphology Identification Data Aggregated over the Satellite-era (MIDAS, Wood and Hartmann, 2006; McCoy et al., 2023, updated) was initially developed to distinguish closed cell mesoscale cellular convection (MCC) from open cell MCC in Sc decks in subtropical subsidence regions. A third cloud type, disorganized but cellular, identifies shallow ocean clouds that do not readily fit into the other two categories. These morphologies will be referred to as MIDAS closed, MIDAS open, and MIDAS disorganized throughout this manuscript.

The MIDAS routine was trained by human observers classifying visible MODIS imagery. These classifiers were then used to train a neural network, which used the mean and spatial variability in the MODIS 6.1 L2 liquid water path (LWP, King et al., 1997; Platnick et al., 2003) field within 256km square boxes to produce classifications for 2003 through 2018. These boxes are spaced 128km apart, allowing for 50% overlap between neighboring boxes. Observations are screened for ice in that LWP is required for classifications. Classified scenes are rejected if the cloud top temperature – SST difference is greater than 30K or if the cloud top is shown to be a majority ice water instead of liquid. Scenes are also rejected if the SST is below 275K. Additional filtration is done to remove the distorting effects of excessive sun glint near the swath center.

### 2.1.2 SGFF

The Sugar, Gravel, Fish, and Flowers (SGFF, Stevens et al., 2019; Schulz et al., 2021) classifications were first developed to distinguish large patches of organized cloud structures in the North Atlantic tropical trade winds. Shallow suppressed Cu cloud scenes are named *Sugar*, while more developed and aggregated shallow convective scenes are named *Gravel*. More stratiform scenes, with geographically separated, horizontally extensive cloud tops and thicker, frequently raining cloud cores are named *Flowers*. The *Fish* classification is named for extensive "bony" structures of thick clouds, often oriented in tendrils aligned in an east-west direction. *Fish* are often associated with the shallow remnants of cold fronts as they dissipate in the tropical trades (Schulz et al., 2021; Aemisegger et al., 2021).





Classifications were initially made by human observers based on visible MODIS images (Rasp et al., 2020), and these classifications were used to train a neural network to identify morphologies in the North Atlantic for years 2003-2020 based on MODIS Aqua infrared brightness temperatures (Schulz et al., 2021). Classifications are made in variably-sized rectangular 120 (WRT latitude-longitude) boxes, often $10° \times 10°$ or larger. Classified regions are permitted to overlap.

### 2.1.3 MEASURES

The third classifier analyzed here is the Community Record Of Marine low-cloud mesoscale Morphology, developed with the support of the NASA Making Earth Science Data Records for Use in Research Environments (MEaSUREs, Yuan et al., 2020; Mohrmann et al., 2021). This routine, built as a continuation and improvement of the MIDAS classifier originally made 125 by Wood and Hartmann (2006), classifies six morphologies present across multiple climate regimes. Morphologies include: solid stratus, closed and open cell MCC, and disorganized clouds observed predominantly in mid-latitude storm track and subtropical subsidence environment, and clustered Cu and suppressed Cu in the tropical trade winds. These tropical cloud types were developed to improve upon the disorganized morphology produced in the MIDAS dataset, which was not trained to discern cloud structures in the tropics and instead classified most tropical scenes as disorganized.

The MEASURES routine was initially trained by human observers classifying MODIS visible images for ocean regions globally. These classifications were used to train a neural network, which employed MODIS visible imagery along with cloud top height, cloud optical depth, cloud drop effective radius, and a cloud mask (Platnick et al., 2017) to produce morphologies globally. Data are available upon request for a selected number of years, including 2018 used here. Classifications are made within 128km square boxes with no overlap between boxes. Classifications are not made near the swath edge (sensor zenith 135 angle $> 45°$) due to the distorting effects of wide view angles on observed cloud properties (Maddux et al., 2010).

## 2.2 Cloud Properties from Satellites

Cloud properties are gathered concurrently with all classifications in order to assess and compare radiative and physical traits. Concurrent datasets are made possible by the formation flying of numerous sensors in NASAs A-Train polar-orbiting satellite 140 constellation. All data are collected during the day at approximately 13:30 local time along the same swath used to generate the classifications (as MODIS on Aqua is part of the A-Train).

Cloud physical properties, including cloud liquid water path (LWP), ice water path (IWP), cloud optical thickness ($\tau$), and cloud droplet effective radius ($r_e$) are sourced from MODIS Aqua L3 optical properties dataset (King et al., 2003; Platnick et al., 2017) on a $1° \times 1°$ latitude-longitude grid. MODIS cloud LWP and $r_e$ are combined to produce an estimate of cloud 145 droplet concentration ($N_d$), as demonstrated in Possner et al. (2020), based on relationships presented in Boers et al. (2006) and Bennartz (2007). The ratio of optically thin to optically thick cloud cover is derived from MODIS liquid cloud optical thickness histograms, which produce counts of observations within optical thickness bins for all observations within $1° \times 1°$ grid boxes. Clouds with a $\tau$ value of less than 3 are considered optically thin, as defined in Leahy et al. (2012). Cloud Cover data is sourced from MODIS cloud mask (Platnick et al., 2017) on the $1° \times 1°$ L3 grid.



Daily albedo is sourced from the Clouds and the Earth's Radiant Energy System (CERES, Loeb et al., 2018) single scanner footprint daily $1° \times 1°$ dataset (SSF1deg, NASA/LARC/SD/ASDC, 2015) based on retrievals from MODIS Aqua. The SSF1deg dataset offers total scene albedo and cloud-free albedo along with cloud amount. These products can be used to calculate the albedo of the cloudy regions within each $1° \times 1°$ grid box, which is the value most frequently applied here.

      Vertical profiles of cloud frequency associated with each morphology are produced using the vertical feature mask (VFM, 
Vaughan et al., 2004) based on LIDAR retrievals taken by the Cloud-Aerosol Lidar with Orthogonal Polarization (CALIOP) carried aboard the Cloud-Aerosol Lidar and Infrared Pathfinder Satellite Observations (CALIPSO) satellite. The VFM dataset produces observations of: clear air, cloud, aerosol, ocean surface, and a flag for when the beam is fully attenuated. Below 8km, profiles contain data in 30-meter vertical bins spaced 333 meters apart along the satellite ground track. Data are available at higher altitudes at reduced spatial resolution. Only 'clear' and 'cloudy' pixels are studied here.

Rain rate data are sourced from the Advanced Microwave Scanning Radiometer (AMSR2) 89 GHz passive microwave brightness temperatures ($T_b$, JAXA, 2012), tuned to estimate rain rates using co-located CloudSat rain-profile observations (Lebsock and L'Ecuyer, 2011) using the routine developed in Eastman et al. (2019). This routine derives rain rate from $T_b$ by controlling for variability in AMSR/2 column integrated water vapor (Wentz et al., 2014), and ERA5 SST and 10-meter wind speed (Copernicus Climate Change Service, 2017), then comparing CloudSat rain rates to $T_b$ values, which tend to be warmer 
when more liquid precipitation is present. This establishes a mean relationship between $T_b$ and rain rate, which is then applied to the full AMSR swath.

## 3  Results

### 3.1  Geographic Distributions

Classifier output from all three routines is plotted on a MODIS visible satellite image for the same day (January 26, 2018) 
in Figure 1 in order to compare the spatial structures of the routines. Frame 1a shows the three MIDAS classifiers in their 256 km square overlapping boxes. Closed and open cells are identified in the subtropical subsidence region in the eastern Atlantic, and are also seen in smaller amounts in the northwestern region, behind the cold front. In the tropical trades, the central-southern portion of the image, the MIDAS routine classifies nearly all features as disorganized. Frame 1b shows the same image classified by the MEASURES routine which uses non-overlapping grid boxes that are half the size of the MIDAS 
boxes. Similar to MIDAS, closed and open Sc cells are seen in the subtropical region, but clouds in the tropical trade winds are mostly classified as clustered or suppressed Cu. Solid St and closed cell Sc dominate the region behind the cold front, where the MIDAS routine did not classify most clouds. The dissipating, trailing edge of the cold front is classified as solid St. In Frame 1c, the SGFF classifications are only present in the subtropics and tropics, with *Flowers* observed upstream where other routines saw Sc types. *Sugar* is observed where the MEASURES routine observed suppressed Cu, off the northwestern African 
coast. Downstream, the remains of the cold front are classified as *Fish*, and a broad area south of the cold front is classified as *Gravel*, where the MEASURES routine classified a mix of suppressed and clustered Cu.





**Figure 1.** The cloud field from January 26, 2018 with overlayed classifications by a) MIDAS, b) MEASURES, c) SGFF. Image Credit: NASA, MODIS AQUA





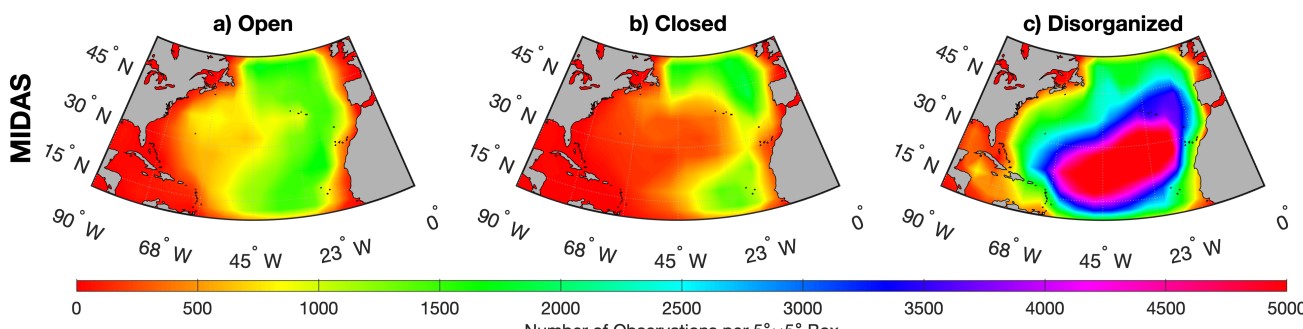

**Figure 2.** The number of $1° \times 1°$ grid boxes classified by MIDAS as a) open MCC, b) closed MCC, and c) disorganized within $5° \times 5°$ grid boxes.

The numbers of observations of each morphology and their geographic distributions are shown in Figures 2-4, where contour maps show how many times morphologies were classified within $5° \times 5°$ latitude-longitude boxes. The grid is aggregated from $1° \times 1°$ in order to show smoother contours, meaning each $1° \times 1°$ box classified within a $5° \times 5°$ grid box counts as a single observation.

MIDAS (Figure 2) open and closed cells are observed less frequently than disorganized and occur in roughly equal amounts in the midlatitude storm track and subtropical subsidence region. Disorganized clouds are extremely common across the entire trade wind belt. This region experiences mean anticyclonic (clockwise) boundary layer wind flow centered over the central Atlantic (Brueck et al., 2015). The peaks in cloud type distributions coincide with this flow. Closed cell MCC transition to open cells further downstream in the mid-latitude storm track and subsidence region. These transition into disorganized clouds even further downstream as clouds are brought into the deeper tropics and trade wind flow toward the Caribbean.

Figure 3 shows the distributions of MEASURES cloud types and adds specificity to the cloud transitions seen in MIDAS associated with the mean anticyclonic Atlantic winds. Furthest upstream in the cold-air-outbreak region, just offshore of the Canadian Maritimes, solid stratus occurrence peaks. Downwind (eastward) of that peak are subsequent distribution peaks in closed MCC, then open MCC, followed by disorganized clouds in the subsidence region offshore of western Europe. In contrast with the MIDAS routine, closed and open MCC are seen less frequently overall, and are classified more frequently in the midlatitude storm track compared to the subsidence region. Disorganized clouds are seen primarily in the eastern Atlantic. Rounding the eastern extreme of the North Atlantic high, clustered Cu occurrence peaks first before suppressed Cu which



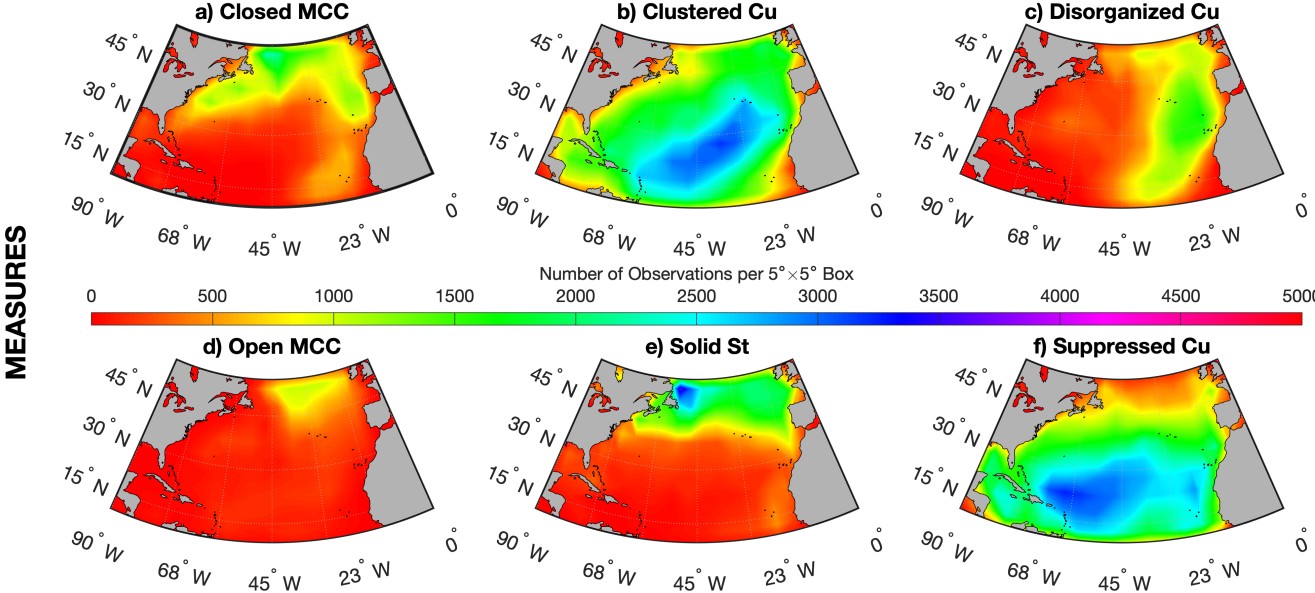

**Figure 3.** The number of $1° \times 1°$ grid boxes classified by MEASURES as a) closed MCC, b) clustered Cu, c) disorganized Cu, d) open MCC, e) solid St, and f) suppressed Cu within $5° \times 5°$ grid boxes.

dominates the downwind trade winds just upwind of the Caribbean. This distribution progression highlights the frequent, Lagrangian morphology transitions that occur as airmasses advect equatorward in the anticyclonic mean flow.

The geographical distributions of the SGFF morphologies are shown in Figure 4 and mainly describe clouds near the tropical belt. *Sugar* is seen most frequently in the upstream trade winds off the coast of Africa with a second region of frequent occurrence nearer the Caribbean. *Gravel* is most frequent closer to the Caribbean, downwind from the *Sugar* maximum. *Flowers* are observed most frequently in a region spanning the subtropical subsidence region and upwind in the tropical belt, where Sc types are generally transitioning to more tropical, Cu cloud types. This is consistent with the more stratiform nature of *Flowers* as observed by Schulz et al. (2021). These distributions suggest *Sugar* cloud types can occur across the trade winds but may frequently form in offshore winds originating over Africa. It is likely these shallower Cu deepen into convective structures akin to *Gravel* and *Flowers* (Narenpitak et al., 2021) as they travel across the trade-winds. *Fish* is the least frequently observed type and is most common in the south-central Atlantic. It should be noted that Schulz et al. (2021) also found *Sugar* frequently occurring adjacent to the ITCZ and its nearby subsidence region.



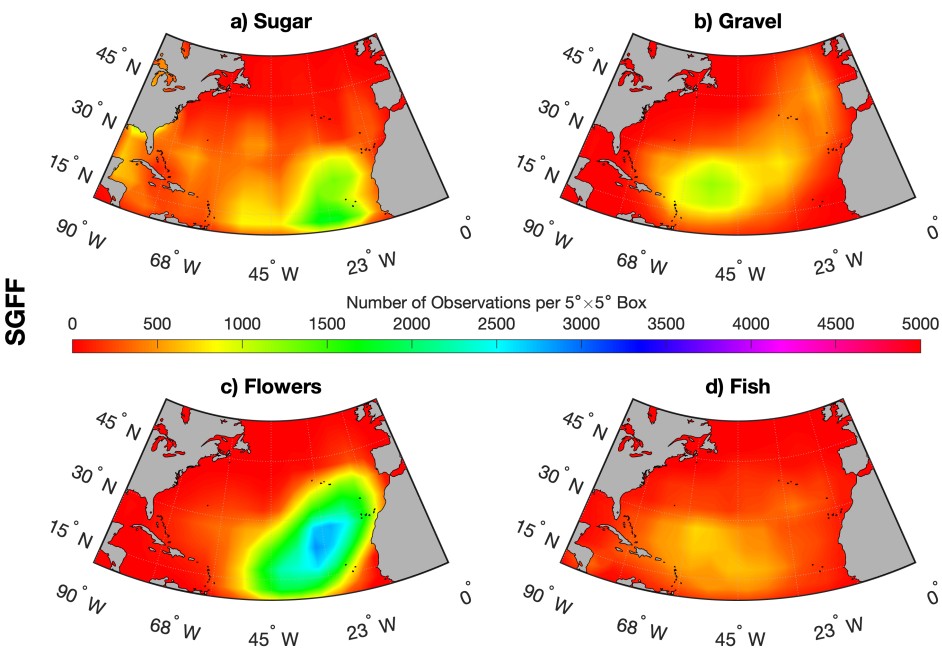

**Figure 4.** The number of $1° \times 1°$ grid boxes classified by SGFF as a) Sugar, b) Gravel, c) Flowers, and d) Fish within $5° \times 5°$ grid boxes.

## 3.2 Overlap Statistics

To assess how the different routines classify the same scenes, Figure 5 illustrates overlap between the morphologies. A $1° \times 1°$ grid box may have multiple classifications assigned by differing routines, or from the same routine due to overlapping observation boxes or a box containing an 'edge' between classifications. To quantify overlap frequency given the varying numbers

of observations of differing morphologies, we show the 'fraction of maximum possible overlaps' between morphologies for all grid boxes in the North Atlantic. The denominator of this fraction is the lower of the two total numbers of observations for any pair of morphologies. The numerator is the number of times the two morphologies are observed in the same $1° \times 1°$ box (same place, same time). The denominator is chosen this way because the smaller total number of observations represents the maximum number of times that the two morphologies can co-occur in time and space. For instance, if there are 1000 total

boxes classified as MIDAS open cells, and 3000 boxes classified as MEASURES open cells, then only 1000 boxes can possibly overlap. If, in this case, 500 boxes are actually observed to overlap, then the fraction of maximum possible overlaps is 0.5.

The fractions of overlap between all types are shown as shades of blue in Figure 5, with darker shades indicating more overlap. This representation allows us to compare scene classification behaviors within and between classifier routines (separated by black lines). MIDAS-classified scenes show the most within-routine overlap of any classifier examined with MEASURES





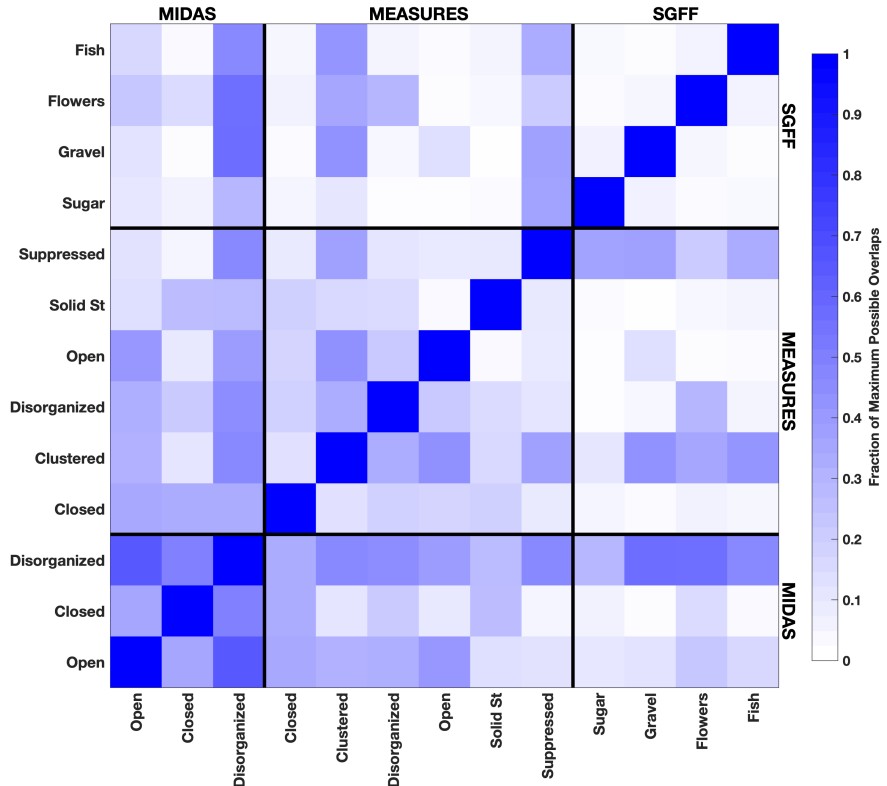

**Figure 5.** The relative amount of overlap seen between any set of two morphology classifications. Shading represents the fraction of maximum possible overlaps: the number of times two morphologies are seen in the same grid box at the same time divided by the lowest of the two total numbers of observations for those two morphologies. Black lines separate classifier routines (labeled at top and right edge). Individual classification labels are marked along the bottom and left edge. Comparison frequencies are repeated across the diagonal for readability.

and SGFF coming second and third, respectively. Within MIDAS, disorganized scenes overlap more with closed or open MCC relative to the less frequent overlap between closed and open MCC. This suggests that edges between closed or open MCC and disorganized scenes are more common than edges between closed and open MCC. Between the MIDAS and MEASURES classifications, open MCC classifications frequently overlap as do MIDAS closed MCC and MEASURES closed MCC and solid St. This suggests broad classification verification for these types between the MIDAS and MEASURES routines. MIDAS

disorganized scenes have the most frequent overlap with other classification routine types, especially those scenes that are not described as closed MCC, solid St, or *Sugar*.

MEASURES clustered scenes overlap with MEASURES open MCC and suppressed Cu scenes. Taken together with the maps from the prior section, a Lagrangian model emerges, where open cells evolve into sparser clustered Cu, which then alternates with suppressed Cu in the tropical trade winds. MEASURES clustered and suppressed scenes overlap with MIDAS

disorganized scenes, showing that the MEASURES routine accomplishes its mission of adding further detail to the expansive



MIDAS disorganized classification. MEASURES clustered scenes overlap with *Gravel* and *Fish*, while MEASURES suppressed scenes overlap more with *Sugar*, *Gravel*, and *Fish*. It is likely that MEASURES is detecting suppressed Cu in gaps between larger features in *Fish* and *Gravel* scenes.

The SGFF classifications show less frequent overlap with one another, but some overlap is apparent between *Sugar-Gravel* and *Fish-Flowers*. *Flowers* overlap most with MIDAS open MCC and disorganized, in addition to MEASURES clustered and disorganized. *Sugar* and MEASURES suppressed Cu show some overlap, as do *Gravel* and clustered Cu, indicating that the two routines are classifying the same scenes as these conceptually similar types.

### 3.3 Morphology and Albedo

In this section, we construct comparisons between various cloud properties for each morphology type to understand the influence morphological organization has on albedo. Generally, we utilize one variable to define quantile bins along the x-axis and report the mean of a second variable in each bin (e.g., shaded lines in Figure 6). The 2-sigma standard error for each bin mean is shown by the line width in the y-direction. To eliminate noise caused by outliers, data plotted in the lines represent the middle 80% quantile (with the upper and lower 10% removed). Large, filled symbols represent the morphology mean x and y behavior. Small, hollow symbols mark the corresponding morphology for each line. We are able to examine inter-morphology (between mean morphology behaviors, compare large symbols) and intra-morphology (within morphology type behaviors, compare shaded lines) to test whether relationships are unique to each type or common to all morphology types.

For each morphology, we find that the scene (all-sky) albedo is both a function of cloud albedo (6a) and cloud amount (6b). The correlation coefficients shown are calculated for the means (large symbols) and describe how much variation between morphologies in mean scene albedo is explained by cloud albedo (6a) and cloud cover (6b). Cloud albedo explains slightly more (98%) of the variability in all-sky albedo compared to cloud cover (90%). Cloud amount and cloud albedo (6c) are also closely related with 86% variance explained.

There is broad agreement in albedo values for similar cloud types seen by different classifiers. For example, MEASURES suppressed Cu and SGFF *Sugar* show nearly equivalent albedos. Albedo tends to be lowest for Cu types and highest for stratiform types, with open cell MCC, disorganized, and clustered Cu albedos in the middle. Stratiform types show far more extensive cloud cover accompanied by a higher albedo compared with Cu types.

The spread between lines in the three plots suggests that the relationships between scene albedo and cloud cover as well as between cloud albedo and cloud amount are unique functions of cloud morphology. This is particularly true for the cloud albedo and cloud amount relationships which exhibit more separation. For instance, the two suppressed Cu cloud types in frame 6c show a significantly less extensive mean cloud amount and a weaker increase in amount when albedo increases relative to stratiform morphologies. Differences are also present in frame 6a, where, again, the suppressed Cu types show a weaker relationship along with *Gravel* and MEASURES open cells. Taken together, these figures show how radiative properties for each cloud morphology are a unique function of cloud cover and cloud albedo.

In addition to the correlations between the mean morphology behaviors, we examine the correlations based on the points used to create the shaded lines in Figures 6a-6c. These coefficients (Table 1) describe how strongly cloud albedo or cloud





**Figure 6.** Morphology relationships (colored lines) for y-axis variables binned into quantiles along the x-axis between a) cloud albedo vs. all-sky albedo, b) cloud amount vs. all-sky albedo, c) cloud albedo vs. cloud amount, and d) cloud amount vs. normalized observation number. Line width in the y-direction represents the 2-σ standard error of the mean. Data for lines exclude the upper and lower 10% quantile bins. Hollow symbols and colors distinguish lines between classifier routine classifications (legend in b) and do not represent any values. In a-b), averages for each morphology are shown as large, corresponding filled symbols. In d), observation number per cloud amount bin are normalized between 0-1 in order to best compare shape.



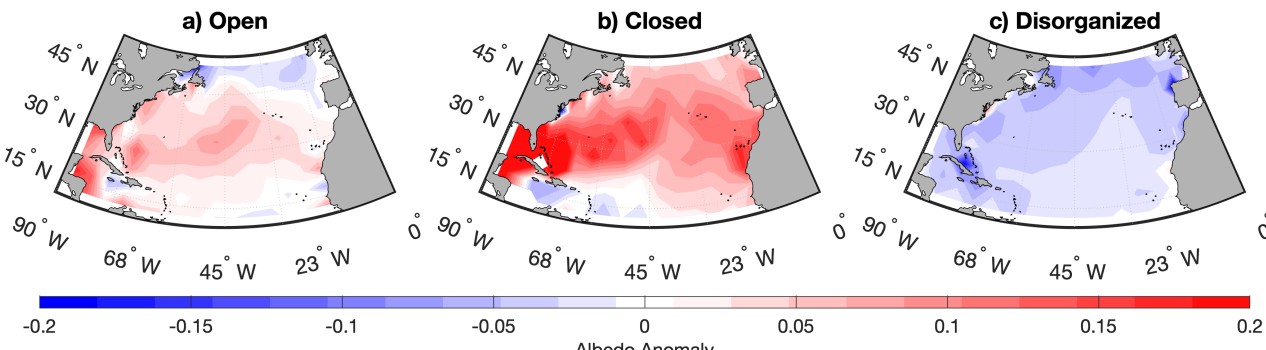

**Figure 7.** Yearly mean of daily albedo anomalies relative to the 100-day running mean for all classifiable low cloud scenes for MIDAS morphologies: a) open MCC, b) closed MCC, and c) disorganized

amount relate to scene albedo for each morphology. For every morphology (with the exception of MEASURES suppressed Cu), cloud albedo is a stronger driver of scene albedo than cloud cover. This is consistent with the mean correlations and implies that, for these morphologies, cloud *reflectivity* may drive all-sky albedo variability more strongly than cloud *amount*. These relationships are generally weaker for the suppressed Cu types, which may indicate difficulty in detecting the larger proportion of optically thin clouds in these predominantly clear scenes.

Figure 6d shows normalized curves comparing the relative number of observations per cloud amount bin for each morphology. These curves show that stratiform types are more frequently observed in cloudy environments, while most of the disorganized or cumuliform types are frequently observed when cloud cover is much lower. However, the distribution of morphology occurrence across the cloud amount space is varied and overlapping, both within and across types, which emphasizes that using cloud amount as a proxy for morphology is not always advantageous.

In Figures 7-9, maps show the geographic distributions of cloud albedo anomaly for each morphology. Albedo anomaly is defined as the daily mean albedo for a specific morphology minus the mean albedo for all sampled low cloud scenes within a $5° \times 5°$ grid box for a 100-day running mean centered on every day. The values shown are annual means. Results show that MEASURES suppressed Cu, *Sugar*, *Gravel*, and MIDAS disorganized scenes have anomalously low albedos throughout our region, while closed cell and solid stratiform clouds almost always have anomalously high albedos. For other types, in-

cluding MIDAS and MEASURES open cell MCC, MEASURES clustered and disorganized Cu, and *Fish* and *Flowers*, the





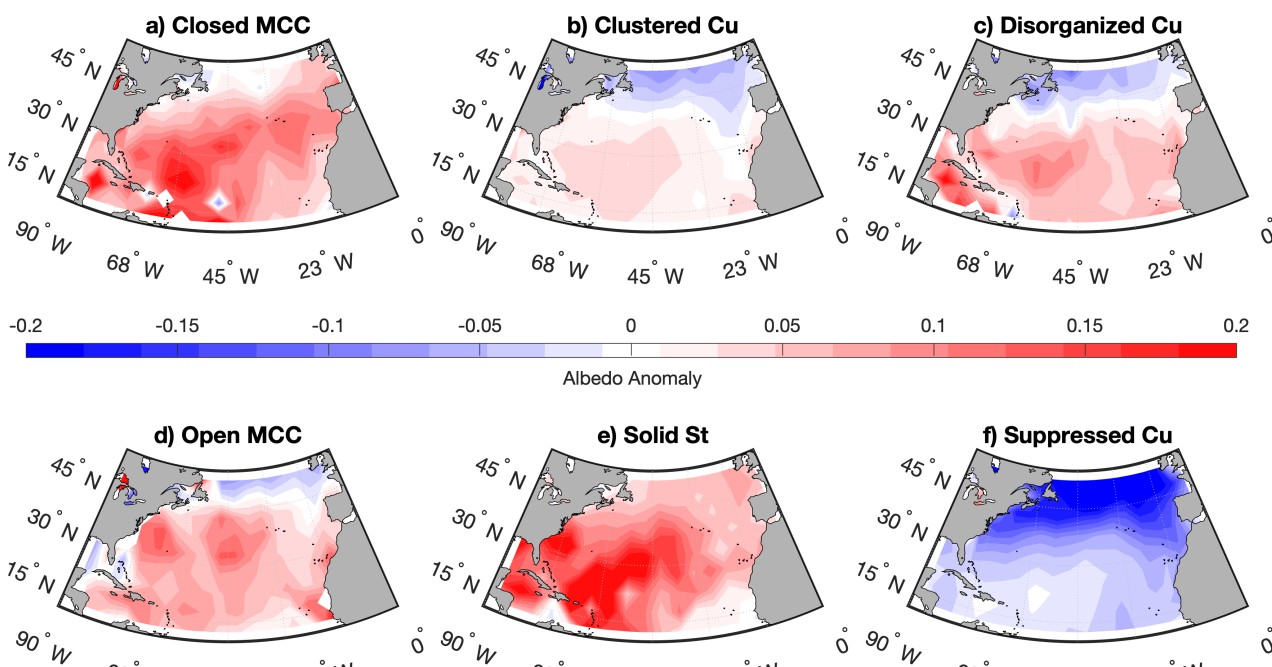

**Figure 8.** As in Figure 7 but for MEASURES morphologies: a) closed MCC, b) clustered Cu, c) disorganized Cu, d) open MCC, e) solid St, and f) suppressed cu

albedo anomaly is negative near the storm track, but positive in more southern regions. This suggests a complex picture where climatologically relevant cloud radiative properties are a function of morphology *and* location.

In Figure 6 and Table 1, cloud albedo was shown to have a profound effect on the all-sky albedo of a cloud scene, greater than cloud amount. This motivates further study into what cloud properties most influence cloud albedo for each morphology. In Figure 10, the relationships between cloud albedo and a number of remotely sensed cloud variables are shown for each morphology using the same method as Figure 6. Data are gathered only for cloud scenes with cloud amounts within 10% of the respective morphology median, which allows us to control for the differing mean cloud amounts associated with the morphologies (sampling the "peaks" relative to cloud amount in 6d). Results were not qualitatively sensitive to changing this threshold to 5% or 20%.

For all morphologies, cloud albedo increases when LWP, $N_d$, $\tau$, and IWP (Fig. 10a,b,c,g, respectively) increase while cloud albedo decreases when the fraction of optically-thin cloud cover increases (Fig. 10h). This final relationship shows the strongest



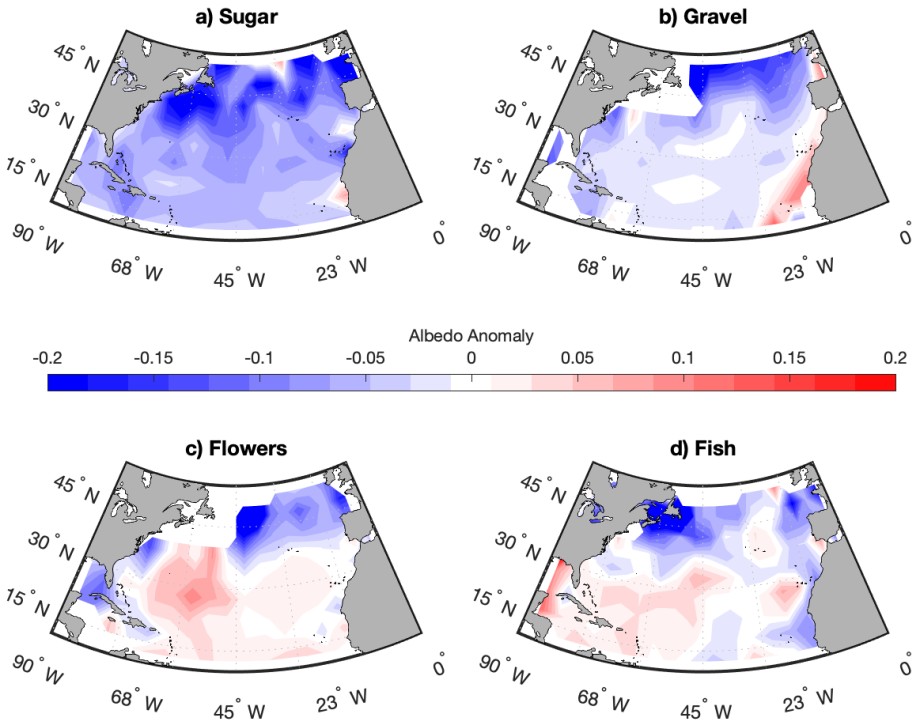

**Figure 9.** As in Figures 7 and 8 but for SGFF morphologies: a) *Sugar*, b) *Gravel*, c) *Flowers*, and d) *Fish*.

correlation, indicating that the fraction of optically-thin cloud cover most strongly explains the cloud albedo variability between morphologies after controlling for cloud amount.

The spread of lines in the y-direction in Figure 10 indicates that there is some degree of cloud albedo equifinality across

morphologies. That is, different morphologies can produce an equivalent cloud albedo despite vastly different cloud properties. A comparison of the plotted lines shows that more cumuliform morphologies such as *Gravel*, disorganized clouds, and open cell MCC have higher values of maximum $\tau$, higher peak rain rates, more ice content and more optically thin clouds compared to the stratiform types for an equivalent cloud albedo. This suggests that the cumuliform morphologies are characterized by thick, raining cores surrounded by optically-thin clouds while stratiform morphologies are much more uniform. We find a

curious difference when comparing mean $\tau$ and peak $\tau$ versus comparing mean rain rate and peak rain rate behaviors. The relationship between $\tau$ and cloud albedo is consistent and positive for both mean and maximum $\tau$ (Fig. 10c,d). However this is not the case for rain rates: mean rain rates are higher for a higher albedo but maximum rain rates are highest for middling





**Figure 10.** Cloud properties: a) in-cloud liquid water path (LWP), b) droplet number concentration ($N_d$), c) mean optical depth ($\tau$), d) maximum optical depth, e) mean rain rate, f) maximum rain rate, g) in-cloud ice water path (IWP), and h) fraction of optically thin cloud features ($\tau < 3$) as a function of cloud albedo. As in Figure 6, symbols show the mean relationship for each classification, lines show the relationship for cloud properties within each morphology for bins of constant cloud albedo, and line width in the y-direction represents the 2-$\sigma$ standard error of the mean.



| Correlation (r) with Scene (All-Sky) Albedo | | | |
| --- | --- | --- | --- |
| | | Cloud Albedo | Cloud Amount |
| MIDAS | Open MCC | 0.92 | 0.66 |
| | Closed MCC | 0.87 | 0.52 |
| | Disorganized | 0.85 | 0.75 |
| MEASURES | Closed MCC | 0.94 | 0.64 |
| | Clustered Cu | 0.83 | 0.77 |
| | Disorganized Cu | 0.89 | 0.73 |
| | Open MCC | 0.91 | 0.75 |
| | Solid St | 0.92 | 0.66 |
| | Suppressed Cu | 0.57 | 0.63 |
| SGFF | Sugar | 0.62 | 0.44 |
| | Gravel | 0.83 | 0.70 |
| | Flowers | 0.79 | 0.71 |
| | Fish | 0.77 | 0.75 |

**Table 1.** Correlations between all-sky albedo and cloud albedo (first column) and between all-sky albedo and cloud amount (second column) for each morphology.

albedos over a broad set of cumuliform morphologies. This nuanced relationship between maximum rain rate and albedo may be associated with heavily precipitating cores surrounded by more optically-thin clouds.

## 3.4 Vertical Profiles and Optical Thickness from CALIPSO

The CALIOP LIDAR aboard CALIPSO provides vertical profiles of cloud tops, and a measure of cloud optical thickness. This section analyses 'cloudy' retrievals in 30m height bins in the lowest 4km of CALIOP LIDAR profiles in classified boxes. Profiles that penetrate the clouds and detect the surface are considered optically thin while fully attenuated profiles are considered optically thick. Optically thick profiles only represent cloud tops, since the true vertical extent of the cloud is unknown due to attenuation of the lidar beam. Profiles that see layered clouds are more complex: occasionally, the lidar sees through upper clouds and attenuates in a lower cloud. Here, layered profiles are split into thin and thick portions. The thin portion represents all clouds that the lidar sees through and the thick portion consists only of the cloud that fully attenuates the beam. Plots in Figure 11 show the fraction of cloudy observations for thin (blue dashed) and thick (solid black) profiles in each height bin divided by the total number of profiles that see cloud for each classification. Combined profiles (sum of thick and thin) are shown as thin black dashed lines. Clear profiles are excluded from the denominator in order to better show and compare the shapes of the cloud profiles. Anomalies for thin and thick profiles, shown respectively as blue and black shaded regions, represent the profiles for each type minus the mean profiles of all types shown in Fig. 11. Frames are arranged to facilitate comparisons between theoretically similar identification types across classifiers.







**Figure 11.** CALIPSO Vertical Feature Mask (VFM) vertical profiles of all (black dashed), optically thin (blue dashed), and thick (black solid) clouds for each morphology. Anomalies relative to the mean profiles in frame (n) are shown as shaded regions (blue for thin, black for thick).





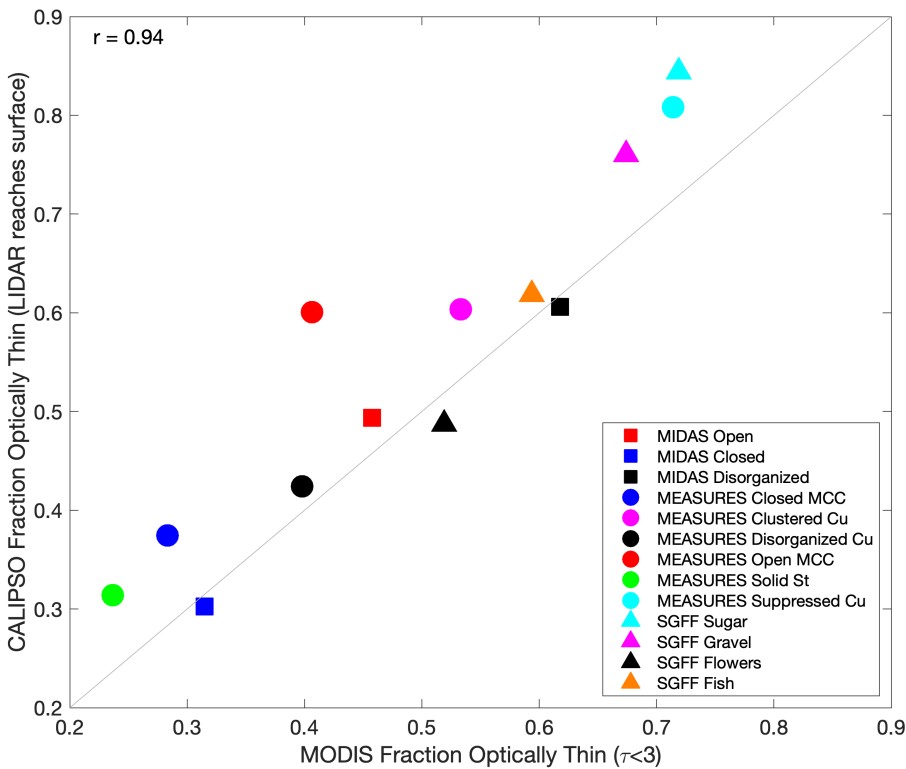

**Figure 12.** Mean optically thin cloud cover fraction as detected by MODIS (# of obs with $\tau < 3$ / total # of cloudy obs) plotted against mean optically thin cloud cover as detected by CALIOP on CALIPSO (# of soundings that see the surface / total # of cloudy soundings).

Results show strong differences in vertical profiles between classifications. Shallower and less vertically distributed types
include suppressed Cu types, closed MCC, MEASURES disorganized, and *Flowers*. Deeper types are *Gravel*, open cells, solid St, and *Fish*. These deeper types tend to rain more heavily (Fig. 10f), except for solid St that are likely associated with weather systems and not convection. Note that, because we are limiting to 4km depth, the *Fish* identifications are likely highlighting the low-level, scud clouds that happen in the vicinity of the larger feature (e.g., Fig. 1). Generally, optically thin clouds tend to be lower in height than surrounding optically thick clouds for all classifications even when accounting for layered scenes.

Comparing absolute and anomaly profiles between similar types classified by different routines, we find similar behaviors for suppressed Cu types (MEASURES vs. *Sugar*) and closed MCC types (MEASURES vs. MIDAS). Some differences are apparent between other theoretically similar types. *Gravel* scenes tend to contain more optically thin and fewer optically thick clouds compared to MEASURES clustered Cu. This may be because more optically thin clouds are present in the larger classification boxes around *Gravel*. Open cells classified by MEASURES contain more clouds, especially optically thin, in the





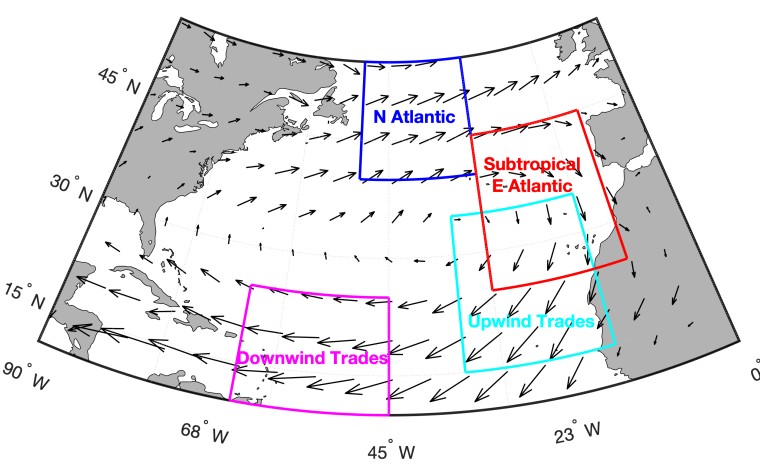

**Figure 13.** The four study regions compared in Figures 14 and 15 for the entire year 2018. Mean wind vectors at 925hPa from ERA5 are shown as arrows, with arrow length scaled by mean wind speed.

upper portions of the profile compared to MIDAS open cells, perhaps owing to the MEASURES open cells disproportionate prevalence in the storm track.

CALIOP provides an independent measure of the fraction of optically thin clouds that can be compared to MODIS. In Figure 12 the number of CALIOP profiles that see the surface divided by the total number of cloudy profiles is plotted against the fraction of cloudy MODIS pixels with $\tau < 3$. Layered CALIOP soundings are considered optically thick here, since an equivalent MODIS observation would be unable to effectively distinguish the layers, and the albedo would be more akin to that of an optically thick scene. We see strong agreement between MODIS and CALIOP measures (88% variance explained), increasing confidence in our assessment of optically thin fractions for the classified types.

### 3.5 Regional Differences

Figure 10 shows significant differences in cloud properties between morphologies while Figures 2-4 show strong differences in their geographic distributions. Given this, we wish to investigate the degree that cloud property differences are influenced by varying geographic distributions separately from morphological differences. To assess this, Figures 14-15 present a comparison of cloud properties in smaller sub-regions that are illustrated in Figure 13. Boxes sample different sub-regions along the anticyclonic flow in the Atlantic, illustrated by mean winds at cloud level (vectors, 925hPa). Boxes in the far North Atlantic



**Figure 14.** Mean cloud properties for each morphology that is commonly observed in the far North Atlantic (x-axis) and subtropical subsidence region (y-axis). Properties include a) Maximum optical depth ($\tau$), b) Cloud droplet effective radius ($r_e$, in place of rain rate), c) Cloud droplet concentration ($N_d$), d) In-cloud liquid water path (LWP), e) Albedo, f) Fraction of optically thin cloud ($\tau<3$), g) Cloud amount, and h) In-cloud ice water path.





**Figure 15.** Mean cloud properties for each morphology that is commonly observed in the far upwind trade winds (x-axis) downwind trade winds (y-axis). Properties include a) Maximum optical depth ($\tau$), b) Rain rate, c) Cloud droplet concentration ($N_d$), d) In-cloud liquid water path (LWP), e) Albedo, f) Fraction of optically thin cloud ($\tau<3$), g) Cloud amount, and h) In-cloud ice water path.



and East Atlantic subsidence regions are chosen to compare stratiform morphologies while boxes in the upwind and downwind
tropical trade winds are chosen in order to compare shallow tropical convective morphologies. Plots compare cloud micro- and
macro-physics, radiative properties, precipitation, and phase. Scatter plots in Figures 14 and 15 present each sub-region on an
axis. A 1:1 line is also shown, where the area-wide North Atlantic mean values for each morphology are shown as a hollow
symbol, in order to compare sub-regional behavior to the entire regional mean.

Figure 14 compares the far North Atlantic with the East Atlantic subsidence region. This effectively contrasts stratiform
morphologies between those that frequently occur in the storm track where cold air outbreaks are common and the Sc cloud
types that commonly occur under a shallow marine inversion. Rain rate data derived from $T_b$ are frequently missing in the
far North Atlantic region due to the presence of ice, so rain rates are replaced by cloud droplet effective radius ($r_e$) in order
to better compare rain characteristics. The far North Atlantic shows a greater maximum $\tau$, marginally fewer but somewhat
bigger cloud drops (likely indicating more rain), greater LWP and IWP, and more cloud cover but fewer optically thin clouds.
These differences suggest a thicker, icier, and rainier cloud environment for all morphologies in the far North Atlantic. As a
consequence of these thicker clouds, the cloud albedo in this region is consistently higher compared to both the entire study
region and the subsidence region in the East Atlantic.

Figure 15 compares warm, shallow convective morphologies between upwind and downwind locations in the tropical trade
winds (note axes changes from Fig. 14). Morphologies observed upwind have lower peak rain rates with more drops and are
marginally cloudier with a smaller proportion of optically thin clouds. The cloud LWP is also marginally lower upwind while
ice is minimal in both sub-regions (not unexpected for the trades). The upwind region shows a higher max $\tau$ and albedo,
suggesting that the cleaner (lower $N_d$), more remote, and likely more developed downwind cloud systems are less reflective.
This result hints at the presence of strong Lagrangian development of cloud systems in the trade winds, which may have
significant radiative implications.

Figure 14 and 15 show that the ordering from high-to-low for most cloud properties generally remains consistent between
types and between regions even though mean values may differ significantly between regions. Some of the variability seen
in Figure 10 is likely caused by geographic differences, but a strong morphology-driven variation of cloud properties is still
present after controlling for regionality. Further, Lagrangian development is apparent when looking at upwind/downwind lo-
cations in the trade winds. Finally, the association between rain rates and albedo may be dependent upon morphology or cloud
phase, with more heavily raining, warm tropical clouds being less reflective, while rainier, and possibly icier, scenes in the
mid-latitudes have a higher albedo.

## 4 Discussion

In Figure 6 and Table 1, we show that cloud albedo is at least as strong a predictor of scene albedo as cloud amount. This
was true in explaining albedo variability between morphologies (Fig 6) and within morphologies (Table 1). Figure 10 goes
on to show that the fraction of optically thin cloud cover best predicts the cloud albedo variability between morphologies,
highlighting the importance of cloud optical thickness in climate studies. This result agrees with the results of McCoy et al.



(2023) in showing that differences in cloud optical thickness between MIDAS morphologies drive albedo variation between those morphologies for a constant cloud amount. Here, we can now extend that conclusion by examining more specialized tropical cloud morphology identifications that were previously considered together as one disorganized type.

The radiative importance of cloud optical depth in our present and future climate has been demonstrated in prior work (McCoy et al., 2023; Hu and Stamnes, 2000; Konsta et al., 2022). Using a radiative-convective model, Hu and Stamnes (2000) found that an increase in cloud optical depth was associated with less warming. McCoy et al. (2023) used present-day morphology observations as a basis for calculating the optical depth component of the shortwave cloud feedback (e.g., Zelinka et al., 2012) from shifts in morphology occurrence under extreme climate scenarios. For example, McCoy et al. (2023) show a

local, positive optical depth feedback on SST warming from morphology shifts, which occurred during the 2015-2016 North East Pacific marine heatwave: closed cell MCC was replaced by less cloudy, optically thinner MIDAS disorganized scenes, increasing sunlight on the sea surface. Here, we expanded the specificity of morphology identifications, especially in the tropics, from those used in McCoy et al. (2023) and found some additional variation in behaviors across morphology types. Our results highlight the potential complexity of understanding morphology feedback onto the climate system in present and fu-

ture climates. Identifying the processes involved in development and transitions across these varied morphology types and the sensitivity of these processes to the environment warrants additional study as well.

    A comparison of climate models by Konsta et al. (2022) shows that models fail to reproduce realistic cloud optical depth variability, often producing no optically thin clouds. This contributes to the 'too few, too bright' problem endemic in cloud representation in general circulation models. That study also showed a model failure to reproduce higher cloud optical depths

observed in thicker Sc scenes. Examining parameterized cloud behaviors in the context of morphological classifications may aid in the reproduction of realistic optical thicknesses. Relating morphology occurrence, and their inherent radiative property signatures, to distinct climate regimes (e.g., McCoy et al., 2017; Mohrmann et al., 2021; McCoy et al., 2023) may also be useful in adding nuance to regime based forcing (e.g., Wall et al., 2022) and feedback (e.g., Myers et al., 2021; Zelinka et al., 2022) calculations. In particular, trade-cumulus feedback is still a large source of uncertainty in climate models and

significantly disagrees with observational estimates (e.g., Myers et al., 2021; Vogel et al., 2022), emphasizing the importance of understanding cloud development and radiative impacts in this region.

    Geographic distribution maps and overlap statistics shown in sections 3.1 and 3.2 suggest that Lagrangian transitions between morphologies are common as the mean flow advects clockwise (anti-cyclonically) around the study region. Given the regional, morphology albedo anomalies we found (Figures 7-9), Earth's radiation budget will be modified by shifts in the location

of the climatological average transition regions or other changes in the most common types of transitions occurring in this basin. A few studies have already addressed the drivers of Lagrangian morphology changes in these regions. Narenpitak et al. (2021) demonstrated that moisture convergence and large-scale ascent can drive a *Sugar*-to-*Flowers* transition in the tradewinds. Eastman et al. (2022) found that increased rain driven by strong winds and its accompanying moisture fluxes can drive a closed-to-open MCC transition in the subtropics while warmer surfaces and strong winds were associated with closed-to-

disorganized transitions. Identifying more such mechanisms influencing climatological and Lagrangian morphology transitions



and their sensitivity to environmental changes will be important for understanding how these transitions will modulate present and future energy flows in the climate system.

Our results also motivate future work examining other frequent, radiatively significant transitions that are now apparent from Figure 5. For example, the change between stratiform types and *Flowers*, suppressed Cu evolving into or from clustered

Cu, and *Flowers* evolving into suppressed or clustered Cu. The importance of examining these regionally specific transitions is further emphasized by our finding that a single morphology may present differing radiative characteristics as it undergoes Lagrangian evolution (e.g., along the flow), as shown by clustered Cu or *Gravel* becoming rainier and increasingly optically thin in the downwind trades (Fig. 15). That relationship combined with the heavier rain observed at middling cloud albedo values in Figure 10 for several shallow convective morphologies hints at a precipitation-driven process where heavy rain may

be associated with less reflective clouds for some morphologies. The differences in droplet concentrations (e.g., higher upwind in the trades) in Figures 14 and 15 also motivate a more detailed examination of aerosol influence on morphology radiative properties (e.g., higher albedo upwind) and cloud evolution.

In Leahy et al. (2012), cloud optical thickness was shown to vary inversely to cloud size. Although our study has not specifically studied the sizes of cloudy elements within each morphology, our results generally agree with this inverse relationship.

Morphologies such as suppressed Cu, which consist of many small clouds, contain a greater proportion of optically thin cloud compared to the broad cells associated with closed cell Sc. Objective classification methods have also found that cloud size is one of four important dimensions to consider in separating cloud morphology types, indicating this is a fundamental property of different organization states (Janssens et al., 2021). This motivates future study of cell size, possibly using methods developed by Zhou et al. (2021) and Janssens et al. (2021) to see whether this inverse relationship applies within each morphology,

or only between morphologies. Both Leahy et al. (2012) and Mieslinger et al. (2022) found that small, optically thin clouds are frequently undetected by remote platforms such as MODIS and CALIPSO, causing significant uncertainty in cloud radiative effects and suggesting that optically thin fractions shown here may be underestimated. Advances in observations may aid in detecting these 'hidden' but radiatively significant clouds.

This comparison may aid in establishing a single set of unique morphologies. Given their radiative and physical characteris-

tics, distinct morphologies likely include: solid stratus, closed cell MCC, open cell MCC (though these may present differently in the storm track compared to subsidence regions), aggregated Cu (a combination of *Gravel* and clustered Cu, with the former presenting a deeper, more developed version of the latter), suppressed Cu (as seen similarly by MEASURES and SGFF *Sugar*), *Fish*, and *Flowers*. Overlap statistics for *Fish* and *Flowers* suggest that these are currently or formerly sub-types of clustered Cu and can be alternately described as disorganized. However, their radiative properties appear unique enough to warrant a

distinct classification as do their formation mechanisms (i.e., the larger structures in *Fish* are typically associated with trailing cold fronts (Schulz et al., 2021; Aemisegger et al., 2021)). It will be necessary for future work to converge on this set of morphologies or one like it to avoid endless proliferation of differing cloud types, causing a lack of comparability in studies. New data sources, including improved geostationary satellites, should be used in producing globally focused versions of these identifications in order to maintain a continuing record.





Finally, this work hints at the presence of both equifinality and multifinality in the cloud-climate system. Equifinality, equivalent outcomes born of diverse processes or properties, is demonstrated by the wide range of cloud properties that can be combined to produce the same albedo, depending on cloud morphology. Multifinality, diverse outcomes resulting from similar perterbations, is also present in this system given the unique cloud processes and properties associated with each morphology: A single perturbation to the climate system will likely favor one cloud morphology over another. These concepts motivate future

research to better quantify the diverse cloud processes and radiative characteristics associated with each unique morphology.

## 5    Conclusions

Three supervised machine learning routines (MIDAS, MEASURES, and SGFF) are used to produce thirteen cloud classifications representing distinct morphologies using MODIS Aqua satellite imagery and radiances over the North Atlantic Ocean for the year 2018. Geographic distributions of morphologies vary between classifiers. MIDAS open and closed cell mesoscale

cellular convection (MCC) are most prevalent in midlatitude storm track and eastern subsidence regions, while MIDAS disorganized scenes are most common in the tropical trade winds. MEASURES stratiform cloud types are most common in the midlatitudes, with disorganized clouds more prevalent in the subsidence region. MEASURES Cu types are most common in the tropical trade winds, with clustered Cu peaking upwind, east of suppressed Cu. All four of the morphologies produced by SGFF are common in the tropical trade winds. A study of classifier overlap (when a $1° \times 1°$ grid box is assigned multi-

ple morphologies) finds that MIDAS disorganized clouds overlap with all of the morphologies frequently seen in the tropical trade wind region. This demonstrates that the added specificity of the SGFF and MEASURES routines have added value by separating this expansive category into distinct subsets.

A comparison of CERES-derived albedos shows that cloud albedo and cloud amount both strongly predict the variability in total scene albedo between morphologies. When analyzing albedo variability within each morphology, the scene albedo

was consistently more strongly correlated with cloud albedo than cloud amount. The fraction of optically thin clouds most strongly predicts the mean cloud albedo compared with other physical quantities such as in-cloud liquid water path, in-cloud ice water path, droplet number concentration, maximum optical depth, or rain rate. Different morphologies display unique combinations of these physical variables to achieve a similar cloud albedo. This equifinality complicates our understanding of what controls cloud albedo and highlights the importance of process understanding and the usefulness of a morphology-based

analysis framework. A comparison with the CALIPSO-derived fractions of optically thin cloud cover shows strong agreement with the MODIS-derived fractions, showing robustness of the MODIS optically-thin feature detection method.

Vertical profiles of optically thin and thick cloud cover are produced using CALIPSO LIDAR data. More vertically extensive morphologies include: clustered Cu types (*Gravel* and clustered), solid St, open cell MCC, and, for clouds below 4 km near these features, *Fish*. Shallower morphologies include: closed cell MCC, *Flowers*, and both suppressed Cu and *Sugar* types.

Optically thin features dominated at lower altitudes, nearer cloud base, despite separating multi-layered scenes into thin upper and thick lower portions.



A geographic breakdown shows strong regional differences in radiative and physical properties. Stratiform morphologies are thicker, rainier, and more reflective in the far North Atlantic region where cold air outbreaks commonly occur compared to the subtropical subsidence region. Trade wind Cu types are less rainy, more reflective, and less optically thin in the upwind trades nearer the subsidence region compared to downwind. This suggests there is some Lagrangian evolution occurring within types as well as variability in cloud properties that may be associated with proximity to aerosol sources on land.

Overall, this work describes how a morphology-driven approach to the study of clouds can provide radiatively-important insights about cloud characteristics and evolution, potentially helping us to better encapsulate cloud behaviors in climate models and reduce uncertainty in climate projections. For the wide range of morphological cloud types examined in this study, we find unique relationships between cloud physical properties and radiation. Future work may improve upon this by tying archetypal cloud morphologies to common climate regimes, identifying processes unique to the development and evolution of each morphology, and examining the sensitivity of these processes to environmental conditions.

*Data availability.*  Datasets Include:

MODIS L2 data used to run classifier routines is available at: https://ladsweb.modaps.eosdis.nasa.gov/archive/allData/61/MYD06_L2/ (Platnick et al., 2015)

CERES data used to quantify albedo are available: https://asdc.larc.nasa.gov/project/CERES/CER_SSF1deg-Hour_Aqua-MODIS_Edition4A (NASA/LARC/SD/ASDC, 2015)

MODIS L3 cloud data are available at: https://ladsweb.modaps.eosdis.nasa.gov/archive/allData/61/MYD08_D3 (Platnick et al., 2017)

ERA5 reanalysis data are available at: https://confluence.ecmwf.int/display/CKB/ERA5%3A+data+documentation (Copernicus Climate Change Service, 2017)

CALIPSO VFM data are available at: https://catalog.data.gov/dataset/calipso-lidar-level-2-vertical-feature-mask-vfm-validated-stage-1-v3-41 (Vaughan et al., 2004)

Rain rate data from AMSR2 and CloudSat are available at: https://www.cloudsat.cira.colostate.edu/community-products/warm-rain-rate-estimates-fron (Eastman et al., 2019)

*Author contributions.*  R. Eastman was lead author and created most of the analyses. I. McCoy initiated and originally organized this project, and also reprocessed MIDAS data and contributed to the overlap study. H. Schulz processed SGFF data for additional years. All co-authors aided in focusing and developing the project.



*Competing interests.* The authors have no competing interests.

*Acknowledgements.* We acknowledge our funding sources. Eastman was supported by NASA grant: 80NSSC19K1274. McCoy was supported by the NOAA Climate and Global Change Postdoctoral Fellowship Program, administered by UCAR's Cooperative Programs for the Advancement of Earth System Science (CPAESS) under award NA18NWS4620043B, and by the NOAA cooperative agreements NA17OAR4320101 and NA22OAR4320151. Schulz was funded by the Cooperative Institute for Climate, Ocean, & Ecosystem Studies (CICOES) under NOAA Cooperative Agreement NA20OAR4320271, Contribution No. 2023-1293. Others... Tianle Yuan and Hua Song have generously made processed MEASURES data available for this study and worked with us to further improve the product.



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
