# Peer review of "A Survey of Radiative and Physical Properties of North Atlantic Mesoscale Cloud Morphologies from Multiple Identification Methodologies"

_EGUsphere, 2023_

## Author Comment (AC1)

Summary:
The authors compare three different supervised neural network classifications of low cloud morphologies in the North Atlantic. The geographic distributions, the overlap statistics and the radiative and physical properties of the different morphologies are discussed in detail. The authors find that the all-sky albedo is more strongly correlated to cloud albedo then cloud amount for nearly all morphologies, and that each morphology displays a distinct set of physical characteristics.
I find the paper to be very well-written and a suitable contribution to ACP. The analyses are carefully done and clearly explained. I only have some minor comments that I detail in the following.

*We thank the reviewer for the thoughtful and positive review. Below, we address the main points in their review.*

**Main comments:**

- Some more comparison of MIDAS and MEASURES 'shared' categories: I was expecting that most differences between MIDAS and MEASURES is in the disorganized Cu type, which is distributed over more classes in MEASURES. However, there are very pronounced differences in the Open MCC class for example. I think the authors should analyse and explain these differences in a bit more detail. Fig. 3d for example shows that MEASURES hardly identifies Open MCCs. Do you understand why this is the case?

*The difference likely lies with the contrast in training regions, and possibly the difference in MODIS data used to generate the classification. While MIDAS was trained exclusively in the subtropics, the MEASURES training was done on images sourced globally, including midlatitude storm tracks. This likely produced a slightly different classification, possibly because open cells seen in the storm tracks and associated with cold air outbreaks contain a great deal more ice, more liquid water, and are more reflective (Fig. 14). The additional channels used for MEASURES (3 visible channels, tau, re, CTH, cloud mask; as opposed to LWP for MIDAS) may 'latch' onto characteristics present in the storm track open MCC that are less common in the subtropics. This may motivate a study contrasting open cells in the subtropics to those in midlatitudes to see if similar processes drive cloud organization, or if there are strong enough differences to motivate two types of open cells. This will be mentioned in our results and discussion.*

- 5 shows the overlap of MEASURES with MIDAS Open MCCs, but are there also cases where MIDAS detects Open MCCs but MEASURES doesn't detect anything? And if so, what are the conditions / regions where this occurs? Also in e.g. Fig. 14, Open MCC from the MIDAS and MEASURES classifiers seem to be the furthest away compared to e.g. the closed and disorganized morphologies. Any ideas why this is so?

*If MIDAS is classifying cloud cover, the MEASURES routine will also classify something, given that the overall conditions warrant a classification. A short overlap study done here compares histograms of cloud types seen by MEASURES when MIDAS detects open cells in the storm track*

*compared to the subsidence region. Results show that MIDAS open MCC in the storm track coincides with MEASURES closed MCC, open MCC, clustered Cu, and solid St. This is likely due to the wide MIDAS classification boxes overlapping with the finer detail in the MEASURES boxes for open cells embedded in other cloud fields. This contrasts with the subtropics, where MEASURES instead preferentially classifies clustered Cu, with some disorganized/suppressed Cu when MIDAS classifies open MCC. This may demonstrate that MEASURES open MCC are representative of open MCC specifically in midlatitudes, with more LWP, while MIDAS open cell classifications span multiple regions. Again, this is likely due to the different classification regions during training, and the different channels used causing MEASURES to prefer open cells like those in the storm track.*

*Concerning cloud conditions when MIDAS and MEASURES classify open cells, in both the subtropics and storm track, MEASURES open MCC shows significantly higher mean LWP and lower cloud cover in both regions, suggesting MEASURES is assigning open MCC to more vigorous open cells, more compatible with open MCC in the storm track (as shown in Fig 14).*

*Tables demonstrating this:*

*# MEASURES detections during open MCC detections by MIDAS in the storm tracks:*

| **Closed** | Clustered | Disorganized | Open MCC | Solid St | Suppressed |
|---|---|---|---|---|---|
| **4665** | 4189 | 1808 | 2226 | 3195 | 928 |

*# MEASURES detections during open MCC detections by MIDAS in the subsidence region:*

| Closed | **Clustered** | Disorganized | Open MCC | Solid St | Suppressed |
|---|---|---|---|---|---|
| 2660 | **8980** | 5506 | 601 | 833 | 5730 |

*storm track: for all boxes classified as open MCC by MIDAS or MEASURES*

| Cloud amount | MIDAS LWP MIDAS | Cloud Amount MEASURES | LWP Measures |
|---|---|---|---|
| 85.16% | 154.32 | 81.17% | 212.81 |

*Subsidence: for all boxes classified as open MCC by MIDAS or MEASURES*

| Cloud amount | MIDAS LWP MIDAS | Cloud Amount MEASURES | LWP Measures |
|---|---|---|---|
| 65.34% | 65.216 | 61.95% | 136.79 |

*This comparison will now be discussed in the paper, emphasizing the differences in LWP, and the likely cause being the different training regions and channels/retrievals used.*

- Seasonality & diurnality: As the authors have data for an entire year, I'd find it very interesting to see the seasonal cycle of morphology occurrence, e.g. in the subdomains shown in Figure 13. Likewise, if nighttime morphologies would be available, a few words on the diurnal cycles of the different classifiers would be very interesting.

*Unfortunately, night data are unavailable for MEASURES and MIDAS, which use MODIS optical properties data in their classification routines, which rely on sunlight. A future Lagrangian study may be able to follow parcels that keep their classifications over the course of a full day, with GOES cloud amounts composited to produce average diurnal cloud amounts throughout the diurnal cycle.*

*A seasonal cycle for classifications within the regions will be developed for this work, as suggested here. If the year of data that we use is sufficient for producing a robust set of cycles, it will be included, and if not, we will motivate this work in the discussion.*

- Rain rate data: I'd like to see 1-2 sentence near L166 regarding how well this routine for deriving rain rates works for the low clouds considered in the study. Especially since the authors find a 'curious difference' when comparing mean and peak OD versus rain rate (L305), I wonder whether this isn't related to the way the rain rates are derived.

*We will now add more detailed discussion of the rain rate product. It is possible that this product could be overwhelmed by the heavy rain rates due to full attenuation of the CloudSat radar used to train the AMSR Tb values, which could cause a failure to quantify stronger rain rates in deeper convection. This should not lead to a decline at high (actual) rain rates, though. Instead, it would show a plateau because attenuated CloudSat profiles are assigned a rain rate based on the unattenuated portion of the profile with a flag (in this case a negative sign that we remove) indicating that the actual rate could be higher. The AMSR Tb product uses this same protocol for Tb values outside of the 'trained' range. A comparison with CloudSat data to see if this is seen in both datasets may be possible when more data are available. Unfortunately, for the study period we were constrained to due to data availability, CloudSat observations are lacking for a large portion of 2018. This will now be discussed in the revision, as suggested.*

More specific comments:
- The SGFF morphologies are mostly written in italics in the manuscript, but not the other morphologies. I'd suggest to also write e.g. *Suppressed Cu* in italics.
- L33: Maybe talk about Stratocumulus and Cumulus as archetypal cloud types than rather cloud organizations?
- Paragraphs starting in L45 and L52: I'd suggest to switch the sequence of the two paragraphs, as the three routines are only introduced in L52 but already discussed in L45.
- L158: I didn't fully understand that 'spaced 333m apart along the satellite ground track' refers to a horizontal spatial resolution of 333 m. Maybe rewrite.

- L254f: I am a bit surprised that SGFF doesn't show a lot of within-routine overlap. Previous studies like Vial et al. (2021, https://doi.org/10.1002/qj.4103) mentioned a lot of overlap among SGFF morphologies. What is different here?

*I believe the issue here is that while overlap is somewhat common for SGFF boxes, it is less frequent than for the other types, so shows up less relative to those in the figure. This is because the MIDAS and MEASURES routines classify nearly every box, and the boxes overlap one-another as a byproduct of design for MIDAS (256km, with overlap/oversampling on every edge), so every classified box overlaps with its neighbors; and as a byproduct of our 1x1 grid projection for MEASURES, where a majority of 1x1 boxes probably contain a border between the 128km along-track boxes, so again, most classified 1x1 boxes by MEASURES contain overlap/borders. Comparing these with SGFF shows that SGFF classifiers definitely overlap sometimes, but the majority probably do not, unlike the other two. We will clarify our discussion of overlap/bordering/co-occurrence in the revision, since some of these overlap signals are different due to geometric constraints.*

- Refer to some literature already in the results section: I'd suggest to add a reference to Mieslinger et al. 2022 in L274; a reference to the statement in L279 that "cloud amount as a proxy ...."; and a reference to McCoy et al. (2023) in L298.
- L280ff: I find the conclusion in L286 regarding the complex picture of morphology and location interesting, but wonder whether it needs Fig. 7-9 in the main text for this. Maybe a selection of the most important subplots is enough? There are already a lot of Figures with many panels and I find it hard to digest all the information. So this could be a good point to reduce information.
- L418ff: I don't really see what you mean here, e.g. how we can see the change from stratiform types to Flowers from Fig. 5, and what suppressed Cu evolves into. Please clarify.
- L425f: Maybe good to mention cold pools as a potential driving process in this context.
- L428: This summary of Leahy et al. (2012) is confusing and seems to contradict what is written in the next sentence. Please rewrite.
- L453: perterbations --> perturbations

*These will be corrected in the revision*

Comments on Figures:
- 1: some colors in panel a) differ from the colors of the three categories. Please explain. Also, please enlarge the axis labels and morphology legends.

*The differing colors in 1a may be due to overlapping boxes, where red and blue may combine to make purple, or red and yellow produce orange. We will enlarge the fonts, and mention the overlapping color issue. The colors may also seem obscured because of the varying background colors in the satellite image. We will point this out in the revision.*

- 2-4: I'd suggest to combine all of them in one figure, such that they don't distribute over different pages. I'd also suggest to use a different color scale – for Fish it's not visible easily if we're at the lower or upper end of the range.

*We will modify the color at the top of the scale in order to reduce this confusion. A combined map was produced, which compressed figures 2-4 into a single page, but the detail in the maps was difficult to resolve without zooming in on the figure. Since another reviewer was upset about zooming in on a (larger) Figure, we think it might be best to keep the separate figures for increased legibility, unfortunately.*

- 14 and 15: I find the comparison in these figures very interesting! Suggestions: Change MCC to MIDAS in the figure legends. And add in the caption what filled vs. hollow symbols refer to.

*We will correct the legend and modify the caption, thanks.*

Review of "A Survey of Radiative and Physical Properties of North Atlantic Mesoscale Cloud Morphologies from Multiple Identification Methodologies" by R. Eastman, I. L. McCoy, H. Schulz, and R. Wood (egusphere-2023-2118)

Several recent machine learning methods have been developed to identify different types of mesoscale patterns of low-level clouds. This study compares the patterns types between three methods, their spatial distributions, and their radiative and microphysical properties. As such, it provides a useful bridge between newer and more qualitative approaches to studying cloud phenomena to older and more quantitative approaches. For this reason, I think it is worth publishing, but I have several ideas for improvement.

*We thank the reviewer for their detailed notes, and have responded to their comments below. The revised manuscript will take these comments into account.*

Major comments:
1)  The underlying source data for the pattern identification methods MODIS is imagery, which I believe relies at least in part on a threshold method for detection. Also, I believe some MODIS pixels are labeled as partly cloudy, rendering plane-parallel retrievals of cloud properties questionable. It would be helpful to have a little discussion about how limitations and assumptions going into the MODIS imagery might affect the identification of cloud patterns and characterization of their properties.

*For identifying morphologies, MODIS retrievals used include all pixels, from clear, to partly cloudy, to fully cloudy in order to generate/locate the repeating patterns needed to classify a region.*

*Overall, this is an excellent point, and speaks to the limitations of using satellite data to resolve processes that occur on a sub-grid scale. We address this in responses to the two subcomments below. We will also make a point of adding these issues to the discussion section along with a call for more field work in order to look into potential biases in satellite-retrieved cloud properties associated with each morphology.*

Subcomment A: The first MODIS issue that comes to mind is that grid box cloud fraction and average cloud optical thickness are highly dependent on whether pixels near the threshold of detection or partly cloudy pixels are identified as cloudy or not. If they are included, cloud fraction will be greater but average cloud optical thickness (or cloud albedo) will be smaller. If they are not included, then cloud fraction will be smaller but average cloud optical thickness (or cloud albedo) will be larger. Because pixels near the threshold of detection or partly cloudy pixels have little impact on radiation flux, whether they are included or not has little impact on the total radiative impact from clouds in the grid box, but it can substantially affect whether differences in radiation flux from clouds are attributed to differences in cloud fraction or differences in cloud optical thickness (or cloud albedo). Since some of the main results concern whether cloud albedo or cloud fraction is more important for all-sky albedo variability, I think it

is important to clarify how this depends on assumptions and decisions made about the MODIS source data.

*For grid box cloud fraction, we use MODIS 'Cloud Mask' data, not the optical properties cloud fractions. We will ensure that this is made clear in the revised data section. The Cloud Mask cloud fraction tends to be generous in assigning cloud cover to pixels compared to the optical properties data. Both are likely imperfect in their own ways, and we will now spend some extra time pointing out that satellite retrievals are imperfect regarding partly cloudy scenes. In other words, our conclusions are based on clouds 'as estimated by satellite sensors and algorithms, rather than ground truth'*

*In the parts of the paper where we do use the optical properties data (LWP, Nd, Re) we use a weighted average of partly cloudy and fully cloudy pixels, weighted by the cloud amounts assigned to those two categories, in order to develop averages representative of the entire cloudy scene, not just the cloud cores. This will be clarified in the text, along with a discussion of the limitations of satellite sensors in defining clouds.*

Subcomment B: The second MODIS issue that comes to mind is that retrievals of cloud droplet number and cloud droplet effective radius are most accurate in areas of extensive homogeneous cloud layers and biased in areas of broken cloud and partly cloudy pixels. For this reason, studies often limit characterization of droplet number and size to areas where retrievals are most accurate, but there is reason to believe that these areas are not representative of the scene as a whole. Since some of the main results concern cloud droplet size and implied rain rate, I think it is important to clarify how this depends on assumptions and decisions made about the MODIS source data.

*As mentioned above, we generate a more scene representative picture of cloud properties for each morphology by weighting properties by partly cloudy or fully cloudy area within each box. We will make sure this is clarified in the text, and will ensure that uncertainties caused by these retrieval limitations are highlighted as issues. This is also important motivation for future work to establish ground truths relative to satellite estimates.*

2) The fact that the morphology data are projected onto a 1x1 latitude-longitude grid and that a 1x1 grid box can contain multiple pattern types raises the issue that two pattern types from different methodologies might be identified in the same grid box yet actually be only geographically adjacent with no geographical overlap. How much this happens would depend on the size of the scene classified into pattern types by the various methods and the spatial autocorrelation of pattern types. Additionally, there would be a sampling bias introduced by the fact that latitude-longitude grid boxes are smaller at higher latitudes so that it is less likely that the grid box would contain multiple pattern types. Although consolidation to a 1x1 grid makes comparison with level-3 MODIS and CERES data simpler, it muddles the interpretation of co-occurrence of various pattern types since it is not known for sure whether there is true geographical overlap between pattern types or whether pattern types are geographically

adjacent. Also, there is ambiguity in matching pattern types to level-3 MODIS and CERES grid box properties when there are multiple pattern types in a grid box.

*We will make a point to mention this issue more specifically when discussing/introducing the figure. As described below, proper fixes generate major computational expense with limited returns, but we will now address the grid-box-area issues in the discussion of the work, as well as by scaling composited data by grid box area, so smaller boxes do not have an outsize effect on averages.*

Subcomment A: I think it might be better to use equal-area grid boxes to avoid the sampling bias with latitude, although this makes comparison with level-3 MODIS and CERES data difficult. It may also be useful to investigate the sensitivity of the results to grid box size.

*The grid-box-area issues associated with the grid boxes decreasing in size with latitude do cause a bias. Converting this study entirely to equal-area boxes would be exponentially more computationally expensive, but we can work to reduce the bias in our overlap/co-occurrence numbers and composites by scaling based on relative grid box area, so that smaller boxes are not given outsized influence. This scaling will be described in the data section, and applied to all figures, along with a discussion of the grid box sizing issue.*

Subcomment B: Although perhaps not feasible, I think a better approach would be to go to a much smaller equal area grid box resolution that would in almost all cases be associated with a single pattern type. Then the true geographical co-occurrence of pattern types from two different methods would be known. The co-occurrence of two different types from the same methodology in adjacent grid boxes could also be determined. This approach would result in a less muddled interpretation.

*We will point out that this would be better, but this level of detail far exceeds the level of detail available in the original classifier data. By converting to 1x1 boxes, we are already oversampling the data, which occur on synoptic scales of ~100-500km$^2$, and need to occur in large-scale, repeating clusters in order to be classified. Classifier routines are not built on high enough resolution data to find precise edges in these structures.*

*An examination of many classified boxes shows ambiguous structures within the originally classified footprints, with portions already showing morphology differences from the classification. This is addressed in the reference documentation, where a box is classified as a morphology based on the morphology seen in the majority of the box. There are situations where open or closed MCC inhabit a box classified as only one type, and unfortunately, we cannot perfectly assess which portion is which without inspecting each individually. Without perfectly sampling the classifications to begin with, the proposed solution offers limited benefit at best, but with major computational expenses. We will discuss this limitation in our revision, though, so readers will understand that co-occurrence within a 1x1 box could be caused by actual overlap, adjacent types, or ambiguity due to low data/routine resolution. Future work could address this by utilizing higher resolution data fields.*

Minor Comments:

1) The first paragraph of the abstract is awkward. It would probably be better to split it up into several more conventional sentences.

*This will be addressed in the revision.*

2) Although it would add another figure set, I think it would be helpful to show a representative scene for each of the pattern types rather than require the reader to go back to three papers to see what each of the pattern types looks like.

*We're torn on this, because this information is already published and readily available for any reader of this paper, while including it incurs more heft and expense on our end. We will experiment with a figure that shows this, possibly a modification of Figure 1.*

3) The text size in Fig. 1 and Fig. 5 is very small and is barely readable without zooming in a bit.

*We will address this in the revision.*

4) The blank areas in Fig. 1 suggest that some scenes are not classified into any type of low cloud pattern. Or were they left out purposefully? Note that non-classification is itself is a type.

*In the revision we will emphasize that areas may not be classified if no clouds are present, or if overlying ice clouds obscure the classifying routines. Additionally, SGFF only classifies strongly archetypal scenes, so many areas are unclassified according to that routine.*

5) I think number of observations is not a good unit to use in Figs. 2-4 since it is difficult to directly interpret. I would prefer instead frequency of occurrence of each pattern type, including perhaps the frequency of non-identification of a type, in which the frequency of each pattern type plus non-identification adds up to 100%. This would enable the reader to know the frequency at which a certain cloud type pattern is identified at a particular location over the North Atlantic.

*We will experiment with this design in the revision.*

6) Figs. 2-4 do not have a color scale that is friendly to people with color-impaired vision.

*This color scale will be revised, as mentioned in the response to review 1*

7) I am not sure that "fraction of maximum overlap" is the best way to characterize how often two pattern types from two different are co-identified. It might be more insightful to calculate the frequency of occurrence that type B is identified when type A is already identified, or the frequency of occurrence that type A is identified when type B is already identified. These numbers may not be the same. For example, let's say that one method has stricter criteria for

identifying open cell stratocumulus compared to another method. At a particular location, method 1 open cell Sc might occur on 50 out of 100 days and method 2 open cell Sc might occur on 25 out of 100 days, but always on days on which method 1 open cell Sc occurs. In this case, method 2 open cell Sc occurs 50% of the time when method 1 open cell Sc is already identified, but method 1 open cell Sc occurs 100% of the time when method 2 open cell Sc is already identified. The method of "fraction of maximum overlap" would yield a value of 1 for the above scenario, which is consistent with method 2 open cell Sc always occurring when method 1 open cell Sc is already identified, but it not inform the reader about how much method 1 open cell Sc occurs given that method 2 open cell Sc is already identified. It seems that this might be useful information.

*We will try to build this alternative figure, where one half (cut diagonally) will have the types along the x-axis as the denominator, and the other half will have the y-axis values as the denominator.*

8)  I suppose the large values of fraction of maximum overlap seen between the MIDAS types in Fig. 5 are due to the 50% overlap between neighboring boxes employed by that cloud pattern identification method. In this case, no physical insight can be drawn from that fact since it arises by construction.
9)  I suppose the small values of fraction of maximum overlap seen between the SGFF types in Fig. 5 are due to the very large spatial scale of clouds identified as a single type that is suggested by Fig. 1. In this case, it is less likely that two different types would occur in the same 1x1 grid box and thus possibly overlap. In this case, it is difficult to draw physical insight from the values of fraction of maximum overlap since they appear to be substantially driven by the large spatial scale of SGFF type identification (assuming that Fig. 1 is representative).
10)  If I understand correctly, the overlap in Fig. 5 between pattern types from different methodologies can arise because they both occur in the same exact area within a 1x1 grid box or because they occur in different areas within the same 1x1 grid box. It seems undesirable that "overlap" does not have a unique physical meaning. There is ambiguity about whether different methods are identifying types that are co-located or adjacent.

*Questions 8-10 are addressed in the response to reviewer 1, where we discuss that the sizes of the classified areas, oversampling, and projection onto the 1x1 grid all have some effect on these statistics independent of actual overlap. We will mention this in the revision, so that readers do not assume that these co-occurrence #'s represent a hard truth, but we still want to show how these properties spatially relate to one another.*

11)  It is not clear to me from the method explanation how cloud albedo and cloud amount are matched to types in 1x1 grid boxes. Since a single 1x1 grid box on one day could contain more than one cloud type from the same method, there is not a unique matching between type and cloud albedo and cloud amount associated with that type. If type A was only a small fraction of the 1x1 grid box, the cloud albedo and cloud amount would primarily be caused by type B but nevertheless get averaged into type A, thus muddling the results. It would be better to calculate

cloud albedo and cloud amount only from those 1x1 grid boxes and days in which only one type was identified for a particular method.

*This is a good point. We will try this and see if it produces appreciably different results. At the very least, it is likely to reduce noise.*

12) Does the relative importance of cloud albedo and cloud amount in explaining all-sky albedo depend on thresholds for cloud identification or inclusion of partial cloud pixels in MODIS?

*Since our cloud amount is based on MODIS cloud mask, the optical properties "partial" cloud pixels do not play a role. In general, the cloud mask incorporates pixels that are fully or partially filled. We will point this out in the revision.*

13) In Fig. 6 the thickness of the lines represents the uncertainty of the mean. By this metric, there is not much overlap between cloud types, especially for the cloud albedo vs. cloud amount plot. But how much overlap is there between distributions. Does the statement "Taken together, these figures show how radiative properties for each cloud morphology are a unique function of cloud cover and cloud albedo" apply to individual scenes or only to the mean?

*Definitely to the mean, we will clarify.*

14) It might be useful to show the annual climatology of low cloud albedo in order to put the albedo anomaly plots in context. One thing that is confusing about the albedo anomaly plots is that they do not appear to add up to zero across all the pattern types for a particular methodology. For example, the SGFF albedo anomaly is negative for all types in the midlatitude North Atlantic. Isn't the climatology of low cloud albedo constructed from the four SGFF types? If so, how can they all have an anomaly less than the climatology? Shouldn't some have an anomaly greater than the climatology to balance out?

*The mean albedo is for all low cloud scenes, regardless of whether they were classified by each routine. So for SGFF, which only classifies when it wants to, there are many unclassified cloud scenes that contribute to the mean albedo, but are not classified as SGFF. Additionally, the sum of all maps would not add up to zero without weighting the sum by frequency of occurrence. We will mention this An annual climatology may be somewhat beyond the scope of this manuscript, but could be made available as a supplement.*

15) I wonder how accurate some of the relationships in Fig. 10 are. With broken and scattered cloud fields, is it really possible to accurately retrieve LWP and droplet number concentration?

*We will point out in the revision that these values should be verified with additional field work, since satellite routines, that rely on several assumptions, may not be able to ascertain exact numbers. We do mention that results are weighted by the portions of observations that are PCL versus fully filled.*

16)  "This section analyses 'cloudy' retrievals in 30m height bins in the lowest 4km of CALIOP LIDAR profiles in classified boxes." How is it handled if there are clouds above 4 km elevation?  These clouds may attenuate the signal and cause misidentification of optical thickness of lower clouds if the signal does not reach the surface.

*This could, indeed, cause a problem if the partially-attenuated beam fully attenuates in an otherwise optically thin cloud, thus mis-classifying it as thick. This could be a problem within the lowest 4km as well and is a limitation of this instrument. In the revision, we will mention this, and could also devise an alternative product that only relies on 'clear' soundings above the low clouds.*

17)  "A 1:1 line is also shown, where the area-wide North Atlantic mean values for each morphology are shown as a hollow symbol". This information should be in the caption.

*This will be added to the caption.*

18)  Possibly the information in Figs. 14-15 could be more simply presented in a table or two.

*We could add a table or two as well if necessary, but I have always preferred comparing visually over comparing numbers in a column.*

19)  Line 453: perterbations -> perturbations

*Fixed*

---

## Author Response (AR1)

Summary:
The authors compare three different supervised neural network classifications of low cloud morphologies in the North Atlantic. The geographic distributions, the overlap statistics and the radiative and physical properties of the different morphologies are discussed in detail. The authors find that the all-sky albedo is more strongly correlated to cloud albedo then cloud amount for nearly all morphologies, and that each morphology displays a distinct set of physical characteristics.
I find the paper to be very well-written and a suitable contribution to ACP. The analyses are carefully done and clearly explained. I only have some minor comments that I detail in the following.

*We thank the reviewer for the thoughtful and positive review. Below, we address the main points in their review.*

**Main comments:**

- Some more comparison of MIDAS and MEASURES 'shared' categories: I was expecting that most differences between MIDAS and MEASURES is in the disorganized Cu type, which is distributed over more classes in MEASURES. However, there are very pronounced differences in the Open MCC class for example. I think the authors should analyse and explain these differences in a bit more detail. Fig. 3d for example shows that MEASURES hardly identifies Open MCCs. Do you understand why this is the case?

*We now say in the discussion: "In Figure 5, open cell co-occurrence statistics are peculiar in that MEASURES open cells tend be infrequent when MIDAS open cells are reported, relative to the opposite relationship where MIDAS open cells are more frequent when MEASURES open cells are present. Differing geographic distributions are also present in Figures 2a and 3d, showing that MEASURES open cells are more confined to the storm track where Figure 15d show much higher LWP values. One possible explanation for this difference is the contrast in training regions between MIDAS, which is exclusively trained in the subtropics, and MEASURES, which is trained globally. The inclusion of the storm track in the MEASURES training region may have increased the LWP threshold for the classification of open cells. A brief comparison of mean LWP between MIDAS and MEASURES open cells in Figure 10a shows that MEASURES open cells have consistently higher LWP, consistent with a sensitivity to differing training regions."*

- 5 shows the overlap of MEASURES with MIDAS Open MCCs, but are there also cases where MIDAS detects Open MCCs but MEASURES doesn't detect anything? And if so, what are the conditions / regions where this occurs? Also in e.g. Fig. 14, Open MCC from the MIDAS and MEASURES classifiers seem to be the furthest away compared to e.g. the closed and disorganized morphologies. Any ideas why this is so?

*If MIDAS is classifying cloud cover, the MEASURES routine will also classify something, given that the overall conditions warrant a classification. A short overlap study done here compares histograms of cloud types seen by MEASURES when MIDAS detects open cells in the storm track*

*compared to the subsidence region. Results show that MIDAS open MCC in the storm track coincides with MEASURES closed MCC, open MCC, clustered Cu, and solid St. This is likely due to the wide MIDAS classification boxes overlapping with the finer detail in the MEASURES boxes for open cells embedded in other cloud fields. This contrasts with the subtropics, where MEASURES instead preferentially classifies clustered Cu, with some disorganized/suppressed Cu when MIDAS classifies open MCC. This may demonstrate that MEASURES open MCC are representative of open MCC specifically in midlatitudes, with more LWP, while MIDAS open cell classifications span multiple regions. Again, this is likely due to the different classification regions during training, and the different channels used causing MEASURES to prefer 'juicier' open cells like those in the storm track.*

*Concerning cloud conditions when MIDAS and MEASURES classify open cells, in both the subtropics and storm track, MEASURES open MCC shows significantly higher mean LWP and slightly lower cloud cover in both regions, suggesting MEASURES is assigning open MCC to more vigorous open cells, more compatible with open MCC in the storm track (as shown in Fig 14).*

*Tables demonstrating this:*

*# MEASURES detections during open MCC detections by MIDAS in the storm tracks:*

| **Closed** | Clustered | Disorganized | Open MCC | Solid St | Suppressed |
|---|---|---|---|---|---|
| **4665** | 4189 | 1808 | 2226 | 3195 | 928 |

*# MEASURES detections during open MCC detections by MIDAS in the subsidence region:*

| Closed | **Clustered** | Disorganized | Open MCC | Solid St | Suppressed |
|---|---|---|---|---|---|
| 2660 | **8980** | 5506 | 601 | 833 | 5730 |

*storm track: for all boxes classified as open MCC by MIDAS or MEASURES*

| Cloud amount MIDAS | LWP MIDAS | Cloud Amount MEASURES | LWP Measures |
|---|---|---|---|
| 85.16% | 154.32 | 81.17% | 212.81 |

*Subsidence: for all boxes classified as open MCC by MIDAS or MEASURES*

| Cloud amount MIDAS | LWP MIDAS | Cloud Amount MEASURES | LWP Measures |
|---|---|---|---|
| 65.34% | 65.216 | 61.95% | 136.79 |

*This LWP comparison is now in the discussion section, as shown above in the response to the first bullet point, emphasizing the differences in LWP, and the likely cause being the different training regions and channels/retrievals used.*

- Seasonality & diurnality: As the authors have data for an entire year, I'd find it very interesting to see the seasonal cycle of morphology occurrence, e.g. in the subdomains shown in Figure 13. Likewise, if nighttime morphologies would be available, a few words on the diurnal cycles of the different classifiers would be very interesting.

*Unfortunately, night data are unavailable for MEASURES and MIDAS, which use MODIS optical properties data in their classification routines, which rely on sunlight. A future Lagrangian study may be able to follow parcels that keep their classifications over the course of a full day, with GOES cloud amounts composited to produce average diurnal cloud amounts throughout the diurnal cycle.*

*A seasonal cycle for classifications within the regions was attempted for this work, however the single season of data was not adequate to produce a noise-free plot. Over the course of a single year, the noise produced by the high cloud filter overwhelmed the seasonal cycle of morphology frequency and produced an extremely noisy and non-representative plot. This was true even when using a running mean with a 30-day window. When all datasets have decades of data available, this could be generated as part of a more wide-ranging climatology.*

*We now say in the discussion: "New data sources, including improved geostationary satellites, should be used in producing globally focused versions of these identifications in order to maintain a continuing record, and to establish daily and seasonal climatologies."*

- Rain rate data: I'd like to see 1-2 sentence near L166 regarding how well this routine for deriving rain rates works for the low clouds considered in the study. Especially since the authors find a 'curious difference' when comparing mean and peak OD versus rain rate (L305), I wonder whether this isn't related to the way the rain rates are derived.

*We now say: "The strong resolution of light precipitation by the 89 GHz microwave band and the CloudSat cloud profiling radar used to develop the product allow us to see light rain associated with many of the cloud morphologies studied here. However, retrievals tend to saturate at fairly low rain rates relative to deeper tropical boundary layer convection, providing only a minimum possible rate. This saturation prevents the rain rate product from precisely resolving rates in the heaviest raining cores in the tropics, so rain rates shown here for the most convective morphologies may be underestimated given this limitation."*

More specific comments:
- The SGFF morphologies are mostly written in italics in the manuscript, but not the other morphologies. I'd suggest to also write e.g. *Suppressed Cu* in italics. *This is done*

- L33: Maybe talk about Stratocumulus and Cumulus as archetypal cloud types than rather cloud organizations? *Changed to cloud type.*
- Paragraphs starting in L45 and L52: I'd suggest to switch the sequence of the two paragraphs, as the three routines are only introduced in L52 but already discussed in L45. *These are switched.*
- L158: I didn't fully understand that 'spaced 333m apart along the satellite ground track' refers to a horizontal spatial resolution of 333 m. Maybe rewrite. *This is rewritten to emphasize 'horizontal spatial resolution'*
- L254f: I am a bit surprised that SGFF doesn't show a lot of within-routine overlap. Previous studies like Vial et al. (2021, https://doi.org/10.1002/qj.4103) mentioned a lot of overlap among SGFF morphologies. What is different here?

*I believe the issue here is that while overlap is somewhat common for SGFF boxes, it is less frequent than for the other types, so shows up less relative to those in the figure. This is because the MIDAS and MEASURES routines classify nearly every box. The boxes overlap one-another as a byproduct of design for MIDAS (256km, with overlap on every edge), so every classified box overlaps with its neighbors for that routine by definition. As a byproduct of our 1x1 grid projection for MEASURES, where a majority of 1x1 boxes probably contain a border between the 128km along-track boxes, most classified 1x1 boxes by MEASURES contain overlap/borders. Comparing these with SGFF shows that SGFF classifiers definitely overlap sometimes, but the majority probably do not, unlike the other two. We now clarify our discussion of overlap to emphasize "co-occurrence" in the revision, since some of these overlap signals may represent different geometric relationships due to geometric constraints.*

- Refer to some literature already in the results section: I'd suggest to add a reference to Mieslinger et al. 2022 in L274; a reference to the statement in L279 that "cloud amount as a proxy ...."; and a reference to McCoy et al. (2023) in L298. *The Mieslinger and McCoy references are added. The 279 statement is modified to not need a reference.*
- L280ff: I find the conclusion in L286 regarding the complex picture of morphology and location interesting, but wonder whether it needs Fig. 7-9 in the main text for this. Maybe a selection of the most important subplots is enough? There are already a lot of Figures with many panels and I find it hard to digest all the information. So this could be a good point to reduce information. *We prefer to keep these Figures to strongly emphasize the radiative contrasts between the morphologies, which is one of the main points of the paper.*
- L418ff: I don't really see what you mean here, e.g. how we can see the change from stratiform types to Flowers from Fig. 5, and what suppressed Cu evolves into. Please clarify. *This is rewritten: "Our results also motivate future work examining other frequent, radiatively significant transitions that are now apparent from the relatively frequent co-occurrences suggested in 5. For example, the change between disorganized types and Flowers or suppressed Cu evolving into or from clustered Cu."*
- L425f: Maybe good to mention cold pools as a potential driving process in this context. *We mention this, though it contradicts recent work by Vogel et al. 2021, which is also mentioned.*

- L428: This summary of Leahy et al. (2012) is confusing and seems to contradict what is written in the next sentence. Please rewrite. *Changed to: "the fraction of optically thin cloud cover was shown to vary inversely with cloud size."*
- L453: perterbations --> perturbations *fixed*

Comments on Figures:
- 1: some colors in panel a) differ from the colors of the three categories. Please explain. Also, please enlarge the axis labels and morphology legends.

*The differing colors in 1a may be due to overlapping boxes, where red and blue may combine to make purple, or red and yellow produce orange. We will enlarge the fonts, and now mention the overlapping color issue. The colors may also seem obscured because of the varying background colors in the satellite image. We point this out in the revision: "Colors may deviate from the legend shown if boxes overlap (red overlapping yellow may appear orange), or if the background color of the image changes."*

- 2-4: I'd suggest to combine all of them in one figure, such that they don't distribute over different pages. I'd also suggest to use a different color scale – for Fish it's not visible easily if we're at the lower or upper end of the range.

*The color scale has been modified. We attempted a combined figure, which compressed figures 2-4 into a single page, but the detail in the maps was difficult to resolve without zooming in on the figure. Since another reviewer was upset about zooming in on a (larger) Figure, we think it might be best to keep the separate figures for increased legibility, unfortunately.*

- 14 and 15: I find the comparison in these figures very interesting! Suggestions: Change MCC to MIDAS in the figure legends. And add in the caption what filled vs. hollow symbols refer to.

*The legend and caption are now modified as suggested*

Review of "A Survey of Radiative and Physical Properties of North Atlantic Mesoscale Cloud Morphologies from Multiple Identification Methodologies" by R. Eastman, I. L. McCoy, H. Schulz, and R. Wood (egusphere-2023-2118)

Several recent machine learning methods have been developed to identify different types of mesoscale patterns of low-level clouds. This study compares the patterns types between three methods, their spatial distributions, and their radiative and microphysical properties. As such, it provides a useful bridge between newer and more qualitative approaches to studying cloud phenomena to older and more quantitative approaches. For this reason, I think it is worth publishing, but I have several ideas for improvement.

*We thank the reviewer for their detailed notes, and have responded to their comments below. The revised manuscript will take these comments into account.*

Major comments:
1) The underlying source data for the pattern identification methods MODIS is imagery, which I believe relies at least in part on a threshold method for detection. Also, I believe some MODIS pixels are labeled as partly cloudy, rendering plane-parallel retrievals of cloud properties questionable. It would be helpful to have a little discussion about how limitations and assumptions going into the MODIS imagery might affect the identification of cloud patterns and characterization of their properties.

*This is an excellent point, and speaks to the limitations of using satellite data to resolve processes that occur on a sub-grid scale. We address this in responses to the two subcomments below and in responses to a few minor comments where these issues are again brought up.*

*For identifying morphologies, MODIS retrievals used include all pixels, from clear, to partly cloudy, to cloudy in order to generate/locate the repeating patterns needed to classify a region.*

Subcomment A: The first MODIS issue that comes to mind is that grid box cloud fraction and average cloud optical thickness are highly dependent on whether pixels near the threshold of detection or partly cloudy pixels are identified as cloudy or not. If they are included, cloud fraction will be greater but average cloud optical thickness (or cloud albedo) will be smaller. If they are not included, then cloud fraction will be smaller but average cloud optical thickness (or cloud albedo) will be larger. Because pixels near the threshold of detection or partly cloudy pixels have little impact on radiation flux, whether they are included or not has little impact on the total radiative impact from clouds in the grid box, but it can substantially affect whether differences in radiation flux from clouds are attributed to differences in cloud fraction or differences in cloud optical thickness (or cloud albedo). Since some of the main results concern whether cloud albedo or cloud fraction is more important for all-sky albedo variability, I think it is important to clarify how this depends on assumptions and decisions made about the MODIS source data.

*This is also an excellent point, and we cannot be certain what is truth given the pixelated nature of satellite data. To address this, we now add to the discussion: "The relative importance of albedo and cloud cover in diagnosing the radiative impact of clouds is in-part dependent upon thresholds used in satellite retrievals to identify cloudy or clear scenes. It is possible that a modification of the threshold used to separate cloudy from clear pixels in the MODIS cloud mask employed here could modify our results. Because no perfect truth exists for quantifying satellite-detected clouds, we motivate future work in this area to improve the resolution of cloud cover retrievals."*

Subcomment B: The second MODIS issue that comes to mind is that retrievals of cloud droplet number and cloud droplet effective radius are most accurate in areas of extensive homogeneous cloud layers and biased in areas of broken cloud and partly cloudy pixels. For this reason, studies often limit characterization of droplet number and size to areas where retrievals are most accurate, but there is reason to believe that these areas are not representative of the scene as a whole. Since some of the main results concern cloud droplet size and implied rain rate, I think it is important to clarify how this depends on assumptions and decisions made about the MODIS source data.

*In Figure 10, where we use the optical properties data (LWP, Nd, IWP) we have now also produced a second version of the figure using a weighted average of partly cloudy and fully cloudy pixels, weighted by the cloud amounts assigned to those two categories by MODIS, in order to develop averages representative of the entire cloudy scene, not just the cloud cores (the original Figure 10 just used the 'filled' pixels). This second Figure 10 is shown and discussed in our response to minor comment 15, and shows little to no qualitative difference compared to the original that relies only on 'Filled' MODIS pixels. This suggests that the cloud edge issues likely do not cause significant biases in our results. We also now mention this issue in the discussion of Figure 10 (also shown in the response to comment 15).*

2)  The fact that the morphology data are projected onto a 1x1 latitude-longitude grid and that a 1x1 grid box can contain multiple pattern types raises the issue that two pattern types from different methodologies might be identified in the same grid box yet actually be only geographically adjacent with no geographical overlap. How much this happens would depend on the size of the scene classified into pattern types by the various methods and the spatial autocorrelation of pattern types. Additionally, there would be a sampling bias introduced by the fact that latitude-longitude grid boxes are smaller at higher latitudes so that it is less likely that the grid box would contain multiple pattern types. Although consolidation to a 1x1 grid makes comparison with level-3 MODIS and CERES data simpler, it muddles the interpretation of co-occurrence of various pattern types since it is not known for sure whether there is true geographical overlap between pattern types or whether pattern types are geographically adjacent. Also, there is ambiguity in matching pattern types to level-3 MODIS and CERES grid box properties when there are multiple pattern types in a grid box.

*This comment contains numerous concepts that we will discuss here, or will refer to our answers to similar issues listed in the 'minor comments' section.*

*The question of overlap versus adjacency is difficult to resolve because the original footprints of the classifiers do not exactly match the patterns. A box is classified as a morphology when some fraction of the scene within the box matches that morphologies' behavior, but not necessarily the entire box. This means that we cannot improve precision regarding overlap without going through the data and classifying sub-regions by hand, which is beyond the scope of a survey paper. Instead, we now mention this unavoidable ambiguity and change our wording to 'co-occurrence', and specifically state that it does not imply overlap. This is addressed more in our response to minor questions 8-10.*

*The shrinking of 1x1 grid boxes with latitude is an issue, and we now account for it by weighting boxes by their relative areas, so smaller boxes nearer the poles have less weight relative to larger boxes. This is done for co-occurrence statistics and for the averaging of cloud properties.*

*Additionally, to address the "*ambiguity in matching pattern types to level-3 MODIS and CERES grid box properties when there are multiple pattern types in a grid box*", we have produced a 'pure' subset of morphologies that incorporates only 1x1 boxes classified as one type by each classifier. We have re-made Figures 6 and 10 using this pure subset to see whether a bias exists. The figures are attached below, and are nearly identical to the originals, showing that no bias of this type is present:*

[Figure]

*Figure 6 with area weights to eliminate the grid box size bias and only for 'Pure' classifications, allowing only grid boxes assigned a single morphology by each routine.*

[Figure]

*Figure 10 with area weights to eliminate the grid box size bias and only for 'Pure' classifications, allowing only grid boxes assigned a single morphology by each routine.*

*We have re-made all other figures using this set of 'Pure' classifications and have seen no consistent biases associated with overlapping observations. The 'Pure' data is less numerous, so some plots are noisier with wider error bounds, but otherwise the results are unchanged. We now mention this test in the text, but do not restrict the analysis to the 'Pure' data.*

Subcomment A: I think it might be better to use equal-area grid boxes to avoid the sampling bias with latitude, although this makes comparison with level-3 MODIS and CERES data difficult. It may also be useful to investigate the sensitivity of the results to grid box size.

*The grid-box-area issues associated with the grid boxes decreasing in size with latitude do cause a bias. Converting this study entirely to equal-area boxes would be exponentially more computationally expensive, but we do now reduce the bias in our overlap/co-occurrence numbers and composites by scaling based on relative grid box area, so that smaller boxes are not given outsized influence. This scaling is done by weighting each 1x1 grid box by its relative surface area, and is applied to all relevant figures, along with a discussion of the grid box sizing issue when we introduce the overlap figure.*

Subcomment B: Although perhaps not feasible, I think a better approach would be to go to a much smaller equal area grid box resolution that would in almost all cases be associated with a single pattern type. Then the true geographical co-occurrence of pattern types from two different methods would be known. The co-occurrence of two different types from the same methodology in adjacent grid boxes could also be determined. This approach would result in a less muddled interpretation.

*Unfortunately, this level of detail far exceeds the level of detail available in the original classifier data. By converting to 1x1 boxes, we are already oversampling the data, which occur on synoptic scales of ~100-500km$^2$, and need to occur in large-scale, repeating clusters in order to be classified. Classifier routines are not built on high enough resolution data to find precise edges in these structures.*

*An examination of many classified boxes shows ambiguous structures within the originally classified footprints, with portions already showing morphology differences from the classification. This is addressed in the reference documentation, where a box is classified as a morphology based on the morphology seen in the majority of the box. There are situations where open or closed MCC inhabit a box classified as only one type (even in the original data), and unfortunately, we cannot perfectly assess which portion is which without inspecting each individually. Without perfectly sampling the classifications to begin with, the proposed solution offers limited benefit at best, but with guaranteed major computational expense. We discuss this limitation in the revision, though, so readers will understand that co-occurrence within a 1x1 box could be caused by actual overlap, adjacent types, or ambiguity due to low data/routine resolution. Future work could address this by utilizing higher resolution data fields.*

*We now add to section 3.2: "The exact nature of these co-occurrences cannot be discerned, so values could represent adjacent morphologies, or overlapped classifications." Additionally, we change the name to co-occurrence instead of overlap, to make clear that we cannot be certain that classifications are overlapping.*

Minor Comments:

1)  The first paragraph of the abstract is awkward. It would probably be better to split it up into several more conventional sentences.

*This is broken up now so that the introductory sentence is separate from the introduction of the routines.*

2)  Although it would add another figure set, I think it would be helpful to show a representative scene for each of the pattern types rather than require the reader to go back to three papers to see what each of the pattern types looks like.

*We were torn on this, because this information is already published and readily available for any reader of this paper, while including it incurs more heft and expense on our end. As a compromise, we have doubled the resolution of Figure 1, so that readers can open the higher-res image and identify the differing structures on the enlarged images.*

3)  The text size in Fig. 1 and Fig. 5 is very small and is barely readable without zooming in a bit.

*Figures 1 and 5 now have larger fonts.*

4)  The blank areas in Fig. 1 suggest that some scenes are not classified into any type of low cloud pattern. Or were they left out purposefully? Note that non-classification is itself is a type.

*We now emphasize that areas may not be classified if no clouds are present, or if overlying ice clouds obscure the classifying routines, or if a sun glint filter detects issues. Additionally, SGFF only classifies strongly archetypal scenes, so many areas are unclassified according to that routine. We do not have output from the routines that provides the locations of unclassified boxes, nor the exact reasons for each exclusion, so we will maintain our focus only on the classified regions. In discussing Figure 1, we point out that some areas are unclassified for the aforementioned variety of reasons, and motivate work to expand classifications to more scenes. We now say:*

*"Clouds are unclassified in a few regions for a variety of reasons. If overlying ice clouds are present, or sun glint is interfering with the retrievals, or if the observed patterns do not adequately satisfy any of the criteria for any morphology, then these are left blank. Future work may be able to identify other morphologies or transitions in these gaps, but we restrict the analysis here to only boxes that are classified."*

5)  I think number of observations is not a good unit to use in Figs. 2-4 since it is difficult to directly interpret. I would prefer instead frequency of occurrence of each pattern type, including perhaps the frequency of non-identification of a type, in which the frequency of each pattern type plus non-identification adds up to 100%. This would enable the reader to know the frequency at which a certain cloud type pattern is identified at a particular location over the North Atlantic.

*This has been changed to show absolute frequency instead.*

6)  Figs. 2-4 do not have a color scale that is friendly to people with color-impaired vision.

*The color scale is modified*

7)  I am not sure that "fraction of maximum overlap" is the best way to characterize how often two pattern types from two different are co-identified. It might be more insightful to calculate the frequency of occurrence that type B is identified when type A is already identified, or the frequency of occurrence that type A is identified when type B is already identified. These numbers may not be the same. For example, let's say that one method has stricter criteria for identifying open cell stratocumulus compared to another method. At a particular location, method 1 open cell Sc might occur on 50 out of 100 days and method 2 open cell Sc might occur on 25 out of 100 days, but always on days on which method 1 open cell Sc occurs. In this case, method 2 open cell Sc occurs 50% of the time when method 1 open cell Sc is already identified, but method 1 open cell Sc occurs 100% of the time when method 2 open cell Sc is already identified. The method of "fraction of maximum overlap" would yield a value of 1 for the above scenario, which is consistent with method 2 open cell Sc always occurring when method 1 open cell Sc is already identified, but it not inform the reader about how much method 1 open cell Sc occurs given that method 2 open cell Sc is already identified. It seems that this might be useful information.

*This figure has been re-made this way, and shows an improved picture of co-occurrence. We thank the reviewer for this alternative. Additionally, the values are now scaled by grid box surface area, so smaller boxes no longer have outsize influence.*

8)  I suppose the large values of fraction of maximum overlap seen between the MIDAS types in Fig. 5 are due to the 50% overlap between neighboring boxes employed by that cloud pattern identification method. In this case, no physical insight can be drawn from that fact since it arises by construction.

*We now mention this: "The oversampling of MIDAS scenes is likely responsible for this" ["this" being the frequent co-occurrence].*

9)  I suppose the small values of fraction of maximum overlap seen between the SGFF types in Fig. 5 are due to the very large spatial scale of clouds identified as a single type that is suggested by Fig. 1. In this case, it is less likely that two different types would occur in the same 1x1 grid box and thus possibly overlap. In this case, it is difficult to draw physical insight from the values of fraction of maximum overlap since they appear to be substantially driven by the large spatial scale of SGFF type identification (assuming that Fig. 1 is representative).
10)  If I understand correctly, the overlap in Fig. 5 between pattern types from different methodologies can arise because they both occur in the same exact area within a 1x1 grid box or because they occur in different areas within the same 1x1 grid box. It seems undesirable that

"overlap" does not have a unique physical meaning. There is ambiguity about whether different methods are identifying types that are co-located or adjacent.

*Questions 8-10 are addressed in the response to reviewer 1, where we discuss that the sizes of the classified areas, oversampling, and projection onto the 1x1 grid all have some effect on these statistics independent of actual overlap. We now mention this in the revision, so that readers do not assume that these co-occurrence #'s represent a hard truth, but we still want to show how these properties spatially relate to one another. Overlap has been changed to "co-occurrence", and we now mention this unavoidable ambiguity specifically in section 3.2. Additionally, we have modified figure 5 based on comment 7 in order to clarify how overlap behaves for any type, given the presence of another.*

11) It is not clear to me from the method explanation how cloud albedo and cloud amount are matched to types in 1x1 grid boxes. Since a single 1x1 grid box on one day could contain more than one cloud type from the same method, there is not a unique matching between type and cloud albedo and cloud amount associated with that type. If type A was only a small fraction of the 1x1 grid box, the cloud albedo and cloud amount would primarily be caused by type B but nevertheless get averaged into type A, thus muddling the results. It would be better to calculate cloud albedo and cloud amount only from those 1x1 grid boxes and days in which only one type was identified for a particular method.

[Figure]

*As shown above, this is Figure 6 with area weights to eliminate the grid box size bias and only for 'Pure' classifications, allowing only grid boxes assigned a single morphology by each routine. We now say: "Each figure in this section has been produced by two methods: one way uses all grid boxes assigned a morphology, while the other uses a 'pure' set that only uses boxes assigned a single morphology by each classifier. This is to ensure that no bias is present due to overlap between morphologies. Figures are qualitatively identical regardless of which method is used, showing no bias caused by overlap."*

12) Does the relative importance of cloud albedo and cloud amount in explaining all-sky albedo depend on thresholds for cloud identification or inclusion of partial cloud pixels in MODIS?

*Since our cloud amount is based on MODIS cloud mask, which does not split the retrieval between PCL and Filled, the optical properties partial vs. filled issues do not play a role. In general, the cloud mask incorporates pixels that are fully or partially filled. We mention this threshold issue in the discussion section now, since cloud cover is not really perfectly quantified by the cloud mask.*

13) In Fig. 6 the thickness of the lines represents the uncertainty of the mean. By this metric, there is not much overlap between cloud types, especially for the cloud albedo vs. cloud amount plot. But how much overlap is there between distributions. Does the statement "Taken together, these figures show how radiative properties for each cloud morphology are a unique function of cloud cover and cloud albedo" apply to individual scenes or only to the mean?

*Definitely to the mean, we now clarify: "The spread between lines in the three plots suggests that the relationships between mean scene albedo and mean cloud cover as well as between mean cloud albedo and mean cloud amount are unique functions of cloud morphology"*

14) It might be useful to show the annual climatology of low cloud albedo in order to put the albedo anomaly plots in context. One thing that is confusing about the albedo anomaly plots is that they do not appear to add up to zero across all the pattern types for a particular methodology. For example, the SGFF albedo anomaly is negative for all types in the midlatitude North Atlantic. Isn't the climatology of low cloud albedo constructed from the four SGFF types? If so, how can they all have an anomaly less than the climatology? Shouldn't some have an anomaly greater than the climatology to balance out?

*The mean albedo is for all low cloud scenes, regardless of whether they were classif'ed by each routine. So, for SGFF, which only classifies when it wants to, there are many unclassified cloud scenes that contribute to the mean albedo, but are not classified as SGFF. Additionally, the sum of all maps would not add up to zero without weighting the sum by frequency of occurrence. We will mention this. An annual climatology is somewhat beyond the scope of this manuscript but is an important future goal.*

*We clarify in that section: "Albedo anomaly is defined as the daily mean albedo for a specific morphology minus the mean albedo for all sampled low cloud scenes (from a combination of all*

*routines)* *…". And "Because of differing frequencies of occurrence for each morphology (as shown in Figure 2-4) the sum of all anomalies in 7-9 should not be zero."*

15) I wonder how accurate some of the relationships in Fig. 10 are. With broken and scattered cloud fields, is it really possible to accurately retrieve LWP and droplet number concentration?

*Figure 10 originally included data only for filled pixels, excluding partly cloudy scenes. We have produced a comparable figure that includes PCL and Filled MODIS optical properties data (weighted by respective amounts) for LWP, Nd, and IWP. Results are qualitatively unchanged, so we will continue to use the 'Filled only' approach to avoid concerns about biasing, even though this suggests that broken and scattered scenes are not likely causing major biases in results. This is now mentioned in the results section.*

*We now say in discussing Figure 10: "Figure 10 uses LWP, Nd, and IWP values from 'Filled' MODIS pixels only, excluding cloud edges or any other scenes where pixels are partially filled. This is to avoid biases caused by assumptions used by the retrievals that may not be appropriate in broken cloud scenes. To see whether including or excluding broken scenes could cause a bias, Figure 10 was also produced using weighted averages of 'Filled' and 'Partly Cloudy' pixels, with separate LWP, Nd, and IWP values averaged for portions of each grid box considered 'Filled' or 'Partly Cloudy', then averaged based on the fraction of 'Filled' or 'Partly Cloudy' scenes within each box. This averaged figure was qualitatively unchanged from the original, suggesting that the relationships seen here are unlikely to be biased by scattered or broken cloud scenes."*

*Tau already relied on a weighted average of PCL and Filled, and we will keep it this way to represent the entire cloud scene, and because tau is less likely to be biased in thin/scattered/broken environments since fewer assumptions based on idealized conditions are required for producing the retrieval. Below is the Figure that uses PCL and Filled:*

[Figure]

*Figure 10, but using 'Filled' and 'PCL' data, showing consistency with the original.*

16) "This section analyses 'cloudy' retrievals in 30m height bins in the lowest 4km of CALIOP LIDAR profiles in classified boxes." How is it handled if there are clouds above 4 km elevation? These clouds may attenuate the signal and cause misidentification of optical thickness of lower clouds if the signal does not reach the surface.

*This could, indeed, cause a problem if the partially-attenuated beam fully attenuates in an otherwise optically thin cloud, thus mis-classifying it as thick. This could be an issue for multi-layered clouds within the lowest 4km as well. It is unfortunately a limitation of this instrument. As a test, we have produced a high-cloud filtered version of Figure 11 that only incorporates scenes with no cloud cover over 4km. The figure is shown below and is qualitatively similar to the original figure, showing no systematic bias. We now mention this filtered version in the discussion of Figure 11.*

[Figure]

Fraction: # of Observations per Height Bin / Total # of Cloudy Profiles

*Figure identical to Fig. 11, but only for scenes with no overlying cloud.*

*We now say in section 3.4: "Since overlying high clouds could potentially attenuate the lidar beam, this figure was also created only for profiles with no cloud cover above 4km. This filtered subset of our data produced qualitatively comparable results, showing no bias caused by high clouds."*

17) "A 1:1 line is also shown, where the area-wide North Atlantic mean values for each morphology are shown as a hollow symbol". This information should be in the caption.

*This is added to the caption.*

18) Possibly the information in Figs. 14-15 could be more simply presented in a table or two.

*In this case we prefer the visual comparison in order to show the distribution of points relative to 1:1, which is not as obvious in a table.*

19) Line 453: perterbations -> perturbations

*Fixed*